# Causal Discovery via Cholesky Factorization

## Abstract

Discovering the causal relationship via recovering the directed acyclic graph (DAG) structure from the observed data is a challenging combinatorial problem. This paper proposes an extremely fast, easy to implement, and high-performance DAG structure recovering algorithm. The algorithm is based on the Cholesky factorization of the covariance/precision matrix. The time complexity of the algorithm is $\mathcal{O}(p^2 n + p^3)$, where $p$ and $n$ are the numbers of nodes and samples, respectively. Under proper assumptions, we show that our algorithm takes $\mathcal{O}(\log(p))$ or $\mathcal{O}(p)$ samples to exactly recover the DAG structure under proper assumptions. In both time and sample complexities, our algorithm is better than previous algorithms. On synthetic and real-world data sets, our algorithm is significantly faster than previous methods and achieves state-of-the-art performance.

## 1 Introduction

As Schelling had said: "The whole world is thoroughly to caught in reason, but the question is: how did it get caught in the network of reason in the first place?" (Kuhn, 1942; Žižek & von Schelling, 1997), people found that learning the causal inferences between the variables is a fundamental problem and has many applications in biology, machine learning, medicine, and economics. The problem usually is considered as finding a directed acyclic graph (DAG) from an observational joint distribution. Unfortunately, learning the DAG structure from the observations is proved to be an NP-hard problem (Chickering, 1995; Chickering et al., 2004).

The problem is generally formulated as the structural equation model (SEM), where the variable of a child node is a function of its parents with additional noises. Depending on the types of functions (linear or non-linear) and noises (Gaussian, Gumbel, etc.), there are several SEM families, e.g., Spirtes et al. (2000); Geiger & Heckerman (1994); Shimizu et al. (2006). In general, the graph can be identified from the joint distribution only up to Markov equivalence classes. Zhang & Hyvarinen (2012); Peters et al. (2014); Peters & Bühlmann (2014); Gao et al. (2020) propose several SEM forms that make the graph fully identifiable from the observed data.

Various algorithms had been proposed to deal with the problem. Search-based algorithms (Chickering, 2002; Friedman & Koller, 2003; Ramsey et al., 2017; Tsamardinos et al., 2006; Aragam & Zhou, 2015; Teyssier & Koller, 2005; Ye et al., 2019; Lv et al., 2021) generally adopt a score (e.g., BIC (Peters et al., 2014) score, Cholesky score (Ye et al., 2019), remove-fill score (Squires et al., 2020)) to measure the fitness of different graphs over data and then search over the legal DAG space to find the structure that achieves the highest score. However, exhaustive search over the legal DAG space is infeasible when $p$ is large (e.g., there are $4.1\mathrm{e}^{18}$ DAGs for $p = 10$ (Sloane et al., 2003)). Those algorithms go in quest of a trade-off between the performance and the time complexity.

Since Zheng et al. (2018) proposed an approach that converts the traditional combinatorial optimization problem into a continuous program, many methods (Yu et al., 2019; Lee et al., 2019; Ng et al., 2019a;b; Zheng et al., 2020; Lachapelle et al., 2020; Squires et al., 2020; Zhu et al., 2021) have been proposed. Those algorithms formalize the problem as a data reconstruction task with various differentiable constraints on the DAG adjacent matrix and solve it via the augmented Lagrangian method. These algorithms are able to utilize neural networks to approximate the complicated relations between the features in the observed data and achieve good performances. Recently, reinforcement learning based algorithms (Zhu et al., 2020; Wang et al., 2021) also improved the performance by exploring the possible DAG structure candidates. The algorithms update the parameters of the model

via policy gradient as long as it explored a better DAG structure with a higher reward which measures how well an explored structure meets the requirement of DAG and the observed data.

Topology order search algorithms (TOSA) (Ghoshal & Honorio, 2017; 2018; Chen et al., 2019; Gao et al., 2020; Park, 2020) decompose the DAG learning problem into two phases: (i) Topology order learning via conditional variance of the observed data; (ii) Graph estimation depends on the learned topology order. Those algorithms reduce the computation complexity into polynomial time and are guaranteed to recover the DAG structure under some identifiable assumptions. Our method in this paper is also a topology order search algorithm and it merges the two phases in TOSA into one. In each iteration, it attempts to find a child or a contemporary of the current node. Meanwhile, it also determines the corresponding column vector of the adjacent matrix. The mergence brings three main differences: First, the topology order in TOSA is recovered purely based on the conditional variance of the observed data, whereas our method may also take the sparsity of the adjacent matrix into account; Second, the graph LASSO methods, which are commonly adopted to estimate the graph in the second phase in TOSA, encourage the sparsity of the precision matrix, whereas our method is able to encourage the sparsity of the adjacent matrix; Third, the time complexity is reduced significantly. To be specific, the time complexity of our algorithm is $\mathcal{O}(p^2 n + p^3)$, while the fastest algorithm before is $\mathcal{O}(p^5 n)$ (Park, 2020; Gao et al., 2020). Here $p$ and $n$ are the numbers of nodes and samples, respectively. In addition, under proper assumptions, we show that our algorithm takes $\mathcal{O}(\log(p))$ or $\mathcal{O}(p)$ samples to exactly recover the DAG structure. Compared with previous TOSA algorithms, the sample complexity of our method is much better. Experimental results on synthetic data sets, proteins data sets, and knowledge base data set demonstrate the efficiency and effectiveness of our algorithm. For synthetic data sets, compared with previous baselines, our algorithm improves the performance with a significant margin and at least tens or hundreds of times faster. For the proteins data set, we achieve state-of-the-art performance. For the knowledge base data set, we can observe many reasonable structures of the discovered DAG. Our code is uploaded as supplementary material and will be open-sourced upon the acceptance of this paper.

The rest of this paper is organized as follows. In Section 2, we present our algorithm together with the theoretical analysis. In Section 3, numerical results on synthetic data sets, proteins data set, and knowledge base data set are given. Finally, the paper is concluded in Section 4.

**Notations.** The symbol $\| \cdot \|$ stands for the Euclid norm of a vector or the spectral norm of a matrix. For a vector $\boldsymbol{x} = [\boldsymbol{x}_1, \boldsymbol{x}_2, \ldots, \boldsymbol{x}_p] \in \mathbb{R}^p$, $\| \cdot \|_1$ stands for the $\ell_1$-norm, i.e., $\|\boldsymbol{x}_1\| = \sum_{i=1}^{p} |\boldsymbol{x}_i|$. For a matrix $\boldsymbol{X} = [\boldsymbol{X}_{ij}] \in \mathbb{R}^{m \times n}$, $\| \cdot \|_{2,\infty}$ stands for the two-to-infinity norm, i.e., $\|\boldsymbol{X}\|_{2,\infty} = \max_{1 \le i \le m} \|\boldsymbol{X}_{i,:}\|$; $\| \cdot \|_{\max}$ stands for the max norm, $\|\boldsymbol{X}\|_{\max} = \max_{i,j} |\boldsymbol{X}_{ij}|$.

## 2 CAUSAL DISCOVERY VIA CHOLESKY FACTORIZATION (CDCF)

In this section, we first present some preliminaries on DAG, then motivating our algorithm. Next, the detailed algorithm and theoretical guarantees for the exact recovery of the algorithm are given.

### 2.1 PRELIMINARIES

We assume the observed data is entailed by a DAG $\mathcal{G} = (p, V, E)$, where $p$ is the number of nodes, $V = \{v_1, ..., v_p\}$ and $E = \{(v_i, v_j)|i, j \in \{1, ...p\}\}$ represent the set of nodes and edges, respectively. Each node $v_i$ is corresponding to a random variable $X_i$. The observed data matrix $\boldsymbol{X} = [\mathbf{x}_1, ..., \mathbf{x}_p] \in \mathbb{R}^{n \times p}$ where $\mathbf{x}_i$ is consisting of $n$ i.i.d observations of the random variable $X_i$. The joint distribution of $\boldsymbol{X}$ is $P(\boldsymbol{X}) = \prod_{i=1}^{p} P(X_i|Pa_{\mathcal{G}}(X_i))$, where $Pa_{\mathcal{G}}(X_i) := \{X_j|(v_i, v_j) \in E\}$ is the parents of node $X_i$.

Given $\boldsymbol{X}$, we seek to recover the latent DAG topology structure for the joint probability distribution (Hoyer et al., 2008; Peters et al., 2017). Generally, $\boldsymbol{X}$ is modeled via a structural equation model (SEM) with the form

$$X_i = f_i(Pa_{\mathcal{G}}(X_i)) + N_i, \quad (i = 1, ..., p),$$

where $f_i$ is an arbitrary function representing the relation between $X_i$ and its parents, $N_i$ is the jointly independent noise variable.

In this paper, we focus on the linear SEM defined by

$$X_i = \boldsymbol{X}\mathbf{w}_i + N_i, \quad (i = 1, ..., p),$$

where $\mathbf{w}_i \in \mathbb{R}^p$ is a weighted column vector. Let $\boldsymbol{W} = [\mathbf{w}_1, \ldots, \mathbf{w}_p] \in \mathbb{R}^{p \times p}$ be the weighted adjacency matrix, $\boldsymbol{N} = [\boldsymbol{n}_1, \ldots, \boldsymbol{n}_p] \in \mathbb{R}^{n \times p}$ be an additive independent noise matrix, where $\boldsymbol{n}_i$ is $n$ i.i.d observations following the noise variable $N_i$. Then the linear SEM model can be formulated as

$$\boldsymbol{X} = \boldsymbol{X}\boldsymbol{W} + \boldsymbol{N}. \tag{1}$$

We assume the noise deviation of the child variable is approximately larger than that of its parents (see Theorem 2.1 for details). Following this assumption, a classical identifiable form of SEM is the linear-Gaussian SEM, where all $N_i$ are i.i.d. and homoscedastic (Peters & Bühlmann, 2014).

## 2.2 ALGORITHM MOTIVATION

As proposed in McKay et al. (2003); Nicholson (1975), a graph is DAG if and only if the corresponding weighted adjacent matrix $\boldsymbol{W}$ can be decomposed into

$$\boldsymbol{W} = \boldsymbol{P}\boldsymbol{T}\boldsymbol{P}^{\mathrm{T}}, \tag{2}$$

where $\boldsymbol{P}$ is a permutation matrix, $\boldsymbol{T}$ is a strict upper triangular matrix, i.e., $\boldsymbol{T}_{ij} = 0$ for all $i \leq j$.

We denote the scaled permuted data matrix as $\widehat{\boldsymbol{X}} = \frac{1}{\sqrt{n}}\boldsymbol{X}\boldsymbol{P}$, the scaled permuted noise matrix as $\widehat{\boldsymbol{N}} = \frac{1}{\sqrt{n}}\boldsymbol{N}\boldsymbol{P}$, and the permutation order $[i_1^*, i_2^* \ldots, i_p^*] = [1, 2, \ldots, p]\boldsymbol{P}$. We can rewrite (1) as

$$\widehat{\boldsymbol{X}} = \widehat{\boldsymbol{X}}\boldsymbol{T} + \widehat{\boldsymbol{N}}.$$

Then it follows that

$$\widehat{\boldsymbol{X}} = \widehat{\boldsymbol{N}}(\boldsymbol{I} - \boldsymbol{T})^{-1}. \tag{3}$$

Let

$$\mathbb{E}(\widehat{\boldsymbol{N}}^{\mathrm{T}}\widehat{\boldsymbol{N}}) = \widehat{\boldsymbol{\Sigma}}_*^2 = \widehat{\boldsymbol{\Sigma}}^{\mathrm{T}}\widehat{\boldsymbol{\Sigma}}, \tag{4}$$

where $\widehat{\boldsymbol{\Sigma}}_*^2$ is the covariance matrix of the noise variables, $\widehat{\boldsymbol{\Sigma}}$ is upper triangular – the Cholesky factor of $\widehat{\boldsymbol{\Sigma}}_*^2$. Let the diagonal entries of $\widehat{\boldsymbol{\Sigma}}$ be $\sigma_{i_1^*}^2, \sigma_{i_2^*}^2, \ldots, \sigma_{i_p^*}^2$. We know that $\sigma_{i_k^*}^2$ is the conditional variance of $N_{i_k^*}$.

Now using (3) and (4), we have the covariance matrix of the permuted data:

$$\widehat{\boldsymbol{C}}_* = \mathbb{E}(\widehat{\boldsymbol{X}}^{\mathrm{T}}\widehat{\boldsymbol{X}}) = (\boldsymbol{I} - \boldsymbol{T})^{-\mathrm{T}}\mathbb{E}(\widehat{\boldsymbol{N}}^{\mathrm{T}}\widehat{\boldsymbol{N}})(\boldsymbol{I} - \boldsymbol{T})^{-1} = (\boldsymbol{I} - \boldsymbol{T})^{-\mathrm{T}}\widehat{\boldsymbol{\Sigma}}^{\mathrm{T}}\widehat{\boldsymbol{\Sigma}}(\boldsymbol{I} - \boldsymbol{T})^{-1}. \tag{5}$$

Let $\boldsymbol{L} = (\boldsymbol{I} - \boldsymbol{T})^{-\mathrm{T}}\widehat{\boldsymbol{\Sigma}}^{\mathrm{T}}$, then $\widehat{\boldsymbol{C}}_* = \boldsymbol{L}\boldsymbol{L}^T$, which is the Cholesky factorization of the covariance matrix $\widehat{\boldsymbol{C}}_*$ since $\boldsymbol{L}$ is lower triangular. Furthermore, we can see that the diagonal entries of $\boldsymbol{L}$ are the same as that of $\widehat{\boldsymbol{\Sigma}}$, i.e., $\boldsymbol{L}_{kk} = \sigma_{i_k^*}$, the conditional variances of $X_{i_k^*}$ and $N_{i_k^*}$ are the same.

The task becomes to find the permutation $\boldsymbol{i}^* = [i_1^*, i_2^*, \ldots, i_p^*]$ and an upper triangular matrix $\boldsymbol{U}$ such that $\boldsymbol{U}^{-\mathrm{T}}\boldsymbol{U}^{-1}$ is a good approximation of the empirical estimation of the permuted covariance matrix $\widehat{\boldsymbol{C}} = \frac{1}{n}\boldsymbol{X}_{:,\boldsymbol{i}^*}^{\mathrm{T}}\boldsymbol{X}_{:,\boldsymbol{i}^*}$, and $\boldsymbol{U}$ satisfies some additional constraints, such as the sparsity, etc.

## 2.3 ALGORITHM

We iteratively find the permutation $\boldsymbol{i}$ and calculate $\boldsymbol{U}$ via the Cholesky factorization. Assume that $\boldsymbol{i}_{k-1} = [i_1, \ldots, i_{k-1}]$ and $\boldsymbol{U}_{k-1} = \boldsymbol{U}_{1:k-1,1:k-1}$ are settled, and we have

$$\boldsymbol{C}_{1:k-1,1:k-1} = \frac{1}{n}\boldsymbol{X}_{:,\boldsymbol{i}_{k-1}}^{\mathrm{T}}\boldsymbol{X}_{:,\boldsymbol{i}_{k-1}} + \lambda\boldsymbol{I} = \boldsymbol{U}_{k-1}^{-\mathrm{T}}\boldsymbol{U}_{k-1}^{-1}, \tag{6}$$

where $\lambda > 0$ is a diagonal augmentation parameter which we will give detailed discussion latter. Next, we show how to find $i_k$ and the last column of $\boldsymbol{U}_k$.

For the time being, let us assume $i_k$ is known, we show how to compute the last column of $\boldsymbol{U}_k$. Let $\boldsymbol{U}_k^{-1} = \begin{bmatrix} \boldsymbol{U}_{k-1}^{-1} & \boldsymbol{y}_k \\ 0 & \alpha_k \end{bmatrix}$, then

$$\begin{bmatrix} \boldsymbol{U}_{k-1}^{-1} & \boldsymbol{y}_k \\ 0 & \alpha_k \end{bmatrix}^{\mathrm{T}} \begin{bmatrix} \boldsymbol{U}_{k-1}^{-1} & \boldsymbol{y}_k \\ 0 & \alpha_k \end{bmatrix} = \begin{bmatrix} \boldsymbol{U}_{k-1}^{-\mathrm{T}}\boldsymbol{U}_{k-1}^{-1} & \boldsymbol{U}_{k-1}^{-\mathrm{T}}\boldsymbol{y}_k \\ \boldsymbol{y}_k^{\mathrm{T}}\boldsymbol{U}_{k-1}^{-1} & \alpha_k^2 + \|\boldsymbol{y}_k\|^2 \end{bmatrix} = \frac{1}{n} \begin{bmatrix} \boldsymbol{X}_{:,\boldsymbol{i}_{k-1}}^{\mathrm{T}}\boldsymbol{X}_{:,\boldsymbol{i}_{k-1}} + \lambda I & \boldsymbol{X}_{:,\boldsymbol{i}_{k-1}}^{\mathrm{T}}\boldsymbol{X}_{:,i_k} \\ \boldsymbol{X}_{:,i_k}^{\mathrm{T}}\boldsymbol{X}_{:,\boldsymbol{i}_{k-1}} & \|\boldsymbol{X}_{:,i_k}\|^2 + \lambda \end{bmatrix},$$

---

**Algorithm 1** Causal Discovery via Cholesky Factorization (CDCF)

---

1: **input**: Data matrix $\boldsymbol{X} \in \mathbb{R}^{n \times p}$, Truncate Threshold $\omega > 0$, and tuning parameter $\gamma$.
2: **output**: Adjacent Matrix $\mathbf{A}$.
3: Set $\boldsymbol{i} = [1, 2, \ldots, p]$, $R = \|\boldsymbol{X}\|_{2,\infty}^2$ and $\lambda = \frac{\gamma \log p}{n} R$;
4: Set $\ell = \arg\min \{\|\boldsymbol{X}_{:,i_1}\|, \|\boldsymbol{X}_{:,i_2}\|, \ldots, \|\boldsymbol{X}_{:,i_p}\|\}$;
5: Exchange $i_1$ and $i_\ell$ in $\boldsymbol{i}$; Set $\boldsymbol{U}_1 = \sqrt{\frac{n}{\|\boldsymbol{X}_{:,i_\ell}\|^2 + \lambda}}$;
6: **for** $k = 2, 3, \ldots, p$ **do**
7:    **for** $j = k, k+1, \ldots, p$ **do**
8:      $\boldsymbol{y}_j = \frac{1}{n} \boldsymbol{U}_{k-1}^{\mathrm{T}} \boldsymbol{X}_{:,\boldsymbol{i}_{k-1}}^{\mathrm{T}} \boldsymbol{X}_{:,i_j}$;
9:      $\alpha_j = \sqrt{\frac{1}{n} \|\boldsymbol{X}_{:,i_j}\|^2 + \lambda - \|\boldsymbol{y}_j\|^2}$;
10:    **end for**
11:    **(V)** $\ell = \arg\min_{k \le j \le p} \alpha_j^2$;
       **(S)** $\ell = \arg\min_{k \le j \le p} \|\boldsymbol{U}_{k-1} \boldsymbol{y}_j\|_1$;
       **(VS)** $\ell = \arg\min_{k \le j \le p} \|\boldsymbol{U}_{k-1} \boldsymbol{y}_j\|_1 \sqrt{|\alpha_j^2 - \frac{1}{k-1} \sum_{h=1}^{k-1} \frac{1}{[\boldsymbol{U}_{k-1}]_{hh}^2}|}$;
12:    Exchange $i_k$ and $i_\ell$ in $\boldsymbol{i}$;
13:    Set $\boldsymbol{U}_k = \begin{bmatrix} \boldsymbol{U}_{k-1} & -\frac{1}{\alpha_\ell} \boldsymbol{U}_{k-1} \boldsymbol{y}_\ell \\ 0 & \frac{1}{\alpha_\ell} \end{bmatrix}$;
14: **end for**
15: **return** $\mathbf{A} = [\mathrm{TRIU}(\mathrm{TRUNCATE}(\boldsymbol{U}_p, \omega))]_{\mathrm{REVERSE}(\boldsymbol{i}), \mathrm{REVERSE}(\boldsymbol{i})}$.

---

where the last equality dues to (6). It follows that

$$\boldsymbol{y}_k = \frac{1}{n} \boldsymbol{U}_{k-1}^{\mathrm{T}} \boldsymbol{X}_{:,\boldsymbol{i}_{k-1}}^{\mathrm{T}} \boldsymbol{X}_{:,i_k}, \qquad \alpha_k = \sqrt{\frac{1}{n} \|\boldsymbol{X}_{:,i_k}\|^2 + \lambda - \|\boldsymbol{y}_k\|^2}. \tag{7}$$

And direct calculation gives rise to

$$\boldsymbol{U}_k = \begin{bmatrix} \boldsymbol{U}_{k-1}^{-1} & \boldsymbol{y}_k \\ 0 & \alpha_k \end{bmatrix}^{-1} = \begin{bmatrix} \boldsymbol{U}_{k-1} & -\frac{1}{\alpha_k} \boldsymbol{U}_{k-1} \boldsymbol{y}_k \\ 0 & \frac{1}{\alpha_k} \end{bmatrix}. \tag{8}$$

By (8), once $i_k$ is settled, we can obtain the last column of $\boldsymbol{U}_k$. Our task remains to select $i_k$ from $\{1, \ldots, p\} \setminus \{i_1, \ldots, i_{k-1}\}$. There are several ways to accomplish this task. We propose three criteria to select $i_k$. First, we need to compute $\alpha_j$ and $\boldsymbol{y}_j$ by (7) for all possible $j$ ($i_j \in \{1, \ldots, p\} \setminus \{i_1, \ldots, i_{k-1}\}$). Then we select $i_k$ according to one of the following criteria:

**(V)** $i_k = \arg\min_{k \le j \le p} \alpha_j^2$. Under the assumption that the noise variance of the child variable is approximately larger than that of its parents, it is reasonable/natural to select the index that has the lowest estimation of the noise variance. This criterion is guaranteed to find the correct permutation $\boldsymbol{i}^*$ with high probability, which is shown in Section 2.4.

**(S)** $i_k = \arg\min_{k \le j \le p} \|\boldsymbol{U}_{k-1} \boldsymbol{y}_j\|_1$. Using (3) and (6), we know that $\boldsymbol{U}_p$ intends to estimate $(\boldsymbol{I} - \boldsymbol{T}) \widehat{\boldsymbol{\Sigma}}^{-1}$. When the adjacent matrix $\boldsymbol{T}$ is sparse and the noise variables are independent (i.e., $\widehat{\boldsymbol{\Sigma}}$ is diagonal), we would like to select the index that leading to the most sparse column of $\boldsymbol{U}_k$. This criterion is especially useful when the number of samples is small, see Tables B.1, B.2 and B.3 in appendix.

**(VS)** $i_k = \arg\min_{k \le j \le p} \|\boldsymbol{U}_{k-1} \boldsymbol{y}_j\|_1 \sqrt{|\alpha_j^2 - \frac{1}{k-1} \sum_{h=1}^{k-1} \frac{1}{[\boldsymbol{U}_{k-1}]_{hh}^2}|}$. We empirically combine criterion **(V)** and criterion **(S)** together to take both aspects (variance and sparsity) into account. Numerically, we found that this criterion achieves the best performance in real-world data.

The diagonal augmentation trick in (6) is commonly used for an invertible and good conditioned estimation of the covariance matrix (see e.g., (Ledoit & Wolf, 2004)). Such a trick not only ensures that our algorithm does not break down due to the singularity of the sample covariance matrix, but also stabilizes the Cholesky factorization, especially when the sample is insufficient. In addition, by setting $\lambda = \mathcal{O}(\frac{\log p}{n})$, the error bound between the population covariance matrix and the augmented sample covariance matrix does not become worse (see Lemma **??** in the appendix). This trick

significantly improves the ability to recover the DAG, especially when the samples are insufficient, see Tables B.4, B.5 and B.6 in appendix.

The detailed algorithm is summarized in Algorithm 1. Some comments and implementation details follow. Line 4, we select the very initial value $\ell = \arg\min\{\|\boldsymbol{X}_{:,i_1}\|, \|\boldsymbol{X}_{:,i_2}\|, \ldots, \|\boldsymbol{X}_{:,i_p}\|\}$. Line 5, we exchange $i_1$ and $i_\ell$ in $\boldsymbol{i}$ and calculate $\boldsymbol{U}_1 = \sqrt{\frac{n}{\|\boldsymbol{X}_{:,i_\ell}\|^2+\lambda}}$. Lines 6 to 14, we iteratively calculate $\boldsymbol{U}_k$ and update permutation order $\boldsymbol{i}$ until all the indices are settled. Line 15, we truncate $\boldsymbol{U}$, take its strict upper triangular part (denoted by "TRIU") and re-permute the predicted adjacent matrix back to the original order according to the permutation order $\boldsymbol{i}$. Specifically, the truncation is done column-wisely. By (8), the value of $[\boldsymbol{U}_p]_{:,k}$ is inversely proportional to $\alpha_k$. So, for column $k$, we set $\omega_k = \frac{\omega}{\alpha_k}$, and do the truncation: $[\boldsymbol{U}_p]_{ik}$ is set to zero if $|[\boldsymbol{U}_p]_{ik}| < \omega_k$. On output, node $i$ connects to node $j$ in $\mathcal{G}$ if $|\boldsymbol{A}_{ij}| > 0$.

**Time Complexity** Note that we do not have to re-calculate the matrix multiplication of $\boldsymbol{X}_{:,\boldsymbol{i}_{k-1}}^{\mathrm{T}} \boldsymbol{X}_{:,i_j}$ in line 8 since we can calculate $\boldsymbol{C}$ at the cost of $\mathcal{O}(p^2 n)$ at first. Besides, at step $k$, we have already calculate $\boldsymbol{U}_{k-2}^{\mathrm{T}} \boldsymbol{X}_{:,\boldsymbol{i}_{k-1}}^{\mathrm{T}} \boldsymbol{X}_{:,i_j}$ at previous step, we only need to calculate the last entry of $\boldsymbol{y}_j$, which is the inner product between two $k$ dimensional vectors, at the cost of $\mathcal{O}(p)$ in worst case. Overall, the time complexity of CDCF is $\mathcal{O}(p^3 + p^2 n)$. When $n > p$, the complexity becomes $\mathcal{O}(p^2 n)$, which is equivalent to the complexity of calculating the covariance matrix. Additionally, the inner loop (lines 7 to 10) of CDCF can be done in parallel, which makes the algorithm friendly to run on GPU and suitable for large scale calculations.

## 2.4 EXACT DAG STRUCTURE RECOVERY

The following theorem tells that our algorithm is able to recover the DAG exactly with high probability under proper assumptions.

**Theorem 2.1** *Let $\mathbf{x} \in \mathbb{R}^p$ be a zero-mean random vector, $\boldsymbol{C} = \mathbb{E}(\mathbf{x}\mathbf{x}^{\mathrm{T}}) \in \mathbb{R}^{p\times p}$ be the covariance matrix. Let $\mathbf{x}_1, \ldots, \mathbf{x}_n$ be $n$ independent samples, $\widehat{\boldsymbol{C}} = \frac{1}{n}\sum_{k=1}^{n} \mathbf{x}_k\mathbf{x}_k^{\mathrm{T}}$ be the sample covariance estimator. Assume $\|\boldsymbol{C} - \widehat{\boldsymbol{C}}\| \leq \epsilon$ for some $\epsilon > 0$. Denote $\widehat{\boldsymbol{C}}_\lambda = \widehat{\boldsymbol{C}} + \lambda\boldsymbol{I}$, where $\lambda = \mathcal{O}(\epsilon) \geq 0$ is a parameter. Let the Cholesky factorizations of $\boldsymbol{C} = \mathbb{E}\mathbf{x}\mathbf{x}^{\mathrm{T}}$ and $\widehat{\boldsymbol{C}}_\lambda$ be $\boldsymbol{C} = \boldsymbol{L}\boldsymbol{L}^{\mathrm{T}}$ and $\widehat{\boldsymbol{C}}_\lambda = \widehat{\boldsymbol{L}}\widehat{\boldsymbol{L}}^{\mathrm{T}}$, respectively, where $\boldsymbol{L}$ and $\widehat{\boldsymbol{L}}$ are both lower triangular. For the linear SEM model (1), assume (2) and (4), and for $k \in Pa_{\mathcal{G}}(j)$, $\delta = \inf_{k \in Pa_{\mathcal{G}}(j)} \delta_{jk} > 0$, where*

$$\delta_{jk} = \sigma_{i_j^*}^2 + \|\widehat{\boldsymbol{\Sigma}}_n[(\boldsymbol{I} - \boldsymbol{T})^{-1}]_{k:j-1,k}\|^2 - \sigma_{i_k^*}^2.$$

*If $\delta \geq 4(\epsilon + \lambda)$ and $\|\boldsymbol{L}^{-1}\|^2(\epsilon + \lambda) < \frac{3}{4}$, then CDCF-V is able to recover $\boldsymbol{P}$ exactly. In addition, it holds that*

$$\|\mathrm{TRIU}(\boldsymbol{U}_p) - \boldsymbol{T}\|_{\max} \leq 4\|\widehat{\boldsymbol{\Sigma}}_*^{-1}(\boldsymbol{I} - \boldsymbol{T})^{\mathrm{T}}\|_{2,\infty}^2 \|(\boldsymbol{I} - \boldsymbol{T})\widehat{\boldsymbol{\Sigma}}_*^{-\mathrm{T}}\|_{2,\infty}(\epsilon + \lambda),$$

*where $\mathrm{TRIU}(\boldsymbol{U}_p)$ stands for the strictly upper triangular part of $\boldsymbol{U}_p$, $\boldsymbol{U}_p$ is the output of outer loop of Algorithm 1 with criterion (V).*

we know that when $\boldsymbol{T}$ is sparse, we may recover its topology structure by truncating $\boldsymbol{U}_p$.

**Proposition 1** *Let $\boldsymbol{N}_{i,:}$ be independent bounded, or sub-Gaussian, or regular polynomial-tail, then for $n > N(\epsilon)$, it holds $\|\widehat{\boldsymbol{C}}_{xx} - \boldsymbol{C}_{xx}\| \leq \epsilon$, w.h.p. Specifically,*

$$N(\epsilon) \geq C_1 \, \log p \left(\frac{\|(\boldsymbol{I} - \boldsymbol{T})^{-1}\|^2 \|\boldsymbol{C}_{nn}\|}{\epsilon}\right)^2, \qquad \textit{for bounded class;}$$

$$N(\epsilon) \geq C_2 \, p \left(\frac{\|(\boldsymbol{I} - \boldsymbol{T})^{-1}\|^2 \|\boldsymbol{C}_{nn}\|}{\epsilon}\right)^2, \qquad \textit{for the sub-Gaussian class;}$$

$$N(\epsilon) \geq C_3 \, p \left(\frac{\|(\boldsymbol{I} - \boldsymbol{T})^{-1}\|^2 \|\boldsymbol{C}_{nn}\|}{\epsilon}\right)^{2(1+r^{-1})}, \qquad \textit{for the regular polynomial tail class.}$$

The proofs of are provided in the Appendix A. This theorem and proposition also indicates the sample complexity of our algorithm is $\mathcal{O}(p)$. This sample complexity is better than the sample complexities of previous methods, see Table 2.1 for a detailed comparison.

Table 2.1: Sample complexity comparison. The last column represents the $\mathcal{O}$ complexity of the sample number $n$ that makes the algorithm recover the DAG with probability at least $1 - \epsilon$, $p$ is the nodes number, $r$ represents the level of the graph, $d$ is the maximum total degree, $m$ represents the $m$'th moment bounded noise, $g(x) = x/\log x$, $g^{-1}$ exists when $x > 3$ and $g^{-1}(x) > x$.

| Algorithm | Data | Function | Noise Type | Cov($X$) | Sample Complexity |
|---|---|---|---|---|---|
| NPVAR (Gao et al., 2020) | - | (Non)-linear Lip-continuous | - | - | $\mathcal{O}((rp/\epsilon)^{1+p/2})$ |
| EV (Chen et al., 2019) | $n > p$ | Linear | - | $\lambda_{\min} > 0$ | $\mathcal{O}(p^2 \log(p))$ |
|  | $n < p$ | Linear | - | $\lambda_{\min} > 0$ | $\mathcal{O}(q^2 \log(p))$ |
| LISTEN (Ghoshal & Honorio, 2018) | - | Linear | Sub-Gaussian | - | $\mathcal{O}(d^4 \log(p))$ |
|  | - | Linear | Bounded moment | - | $\mathcal{O}(d^4 (p^2)^{1/m})$ |
| US (Park, 2020) | $n > p$ | Linear | Gaussian | $\lambda_{\min} > 0$ $\lambda_{\max} < \infty$ | $g^{-1}(\mathcal{O}(\log(p)))$ |
| CDCF | - | Linear | Sub-Gaussian Polynomial tail | - | $\mathcal{O}(p)$ |

## 3 EXPERIMENTS

In this section, we apply our algorithm to synthetic data sets, proteins data set and knowledge base data set, respectively, to illustrate the efficiency and effectiveness of our algorithm.

### 3.1 LINEAR SEM

We evaluate the proposed methods on simulated graphs from two well-known ensembles of random graph types: Erdös–Rényi (ER) (Gilbert, 1959) and Scale-free (SF) (Barabási & Albert, 1999). The average edge number per node is denoted after the graph type. For example, ER2 represents two edges per node on average. After the graph structure is settled, we assign uniformly random edge weights to obtain a weight matrix $W$. We generate the observation data $X$ from the linear SEM with three noise distributions: Gaussian, Gumbel, Exponential.

We chose our baseline methods as NOTEARS (Zheng et al., 2018), DAG-GNN (Yu et al., 2019), CORL (Wang et al., 2021), NPVAR (Gao et al., 2020), and EQVAR (Chen et al., 2019). Other methods such as PC algorithm (Spirtes et al., 2000), LiNGAM (Shimizu et al., 2006), FGS (Ramsey et al., 2017), MMHC (Tsamardinos et al., 2006), L1OBS (Schmidt et al., 2007), CAM (Bühlmann et al., 2013), RL-BIC2 (Zhu et al., 2020), A*LASSO (Xiang & Kim, 2013), LISTEN (Ghoshal & Honorio, 2018), US (Park, 2020) perform worse than or approximately equal to the selected baselines, and the results can be found in the corresponding papers.

Table 3.1 presents the structural Hamming distance (SHD) of baseline methods and our method on 3000 samples ($n = 3000$). Nodes number $p$ is noted in the first column. Graph type and edge level are noted in the second column. We only report the SHD of different algorithms due to page limitation, and we find that other metrics such as true positive rate (TPR), false discovery rate (FDR), false positive rate (FPR), and F1 score have the similar comparative performance with SHD. We also test bottom-up EQVAR which is equivalent to LISTEN, the result is worse than top-down EQVAR (EV-TD) in this synthesis experiment, so we do not include the result in the table. For $p = 1000$ graphs, we only report the result of EV-TD and CDCF since other algorithms spend too much time (longer than a week) to recover a DAG. We test our algorithms with different variations according to criteria (V, S, VS) introduced in Section 2.3, and with diagonal augmentation trick noted by a "+" as postfix. For example, "CDCF-V" means CDCF with V criterion and $\lambda = 0$, and "CDCF-V+" means CDCF with V criterion and $\lambda = \mathcal{O}(\frac{\log p}{n})$. The implementation details are in the Appendix B. We report the result of CDCF-V+ here, and the results of other CDCF variations can be found in Appendix Table B.4. We run our methods on ten randomly generated graphs and report the mean and variance in the table. Figure 3.1 plots the SHD results tested on 100 nodes graph recovering from different sample sizes. We choose EV-TD and high dimension top down (EV-HTD) as baselines when $p > n$ and $p \leq n$, respectively. We can see from the results, CDCF-V+ achieves significantly better performance comparing with previous baselines.

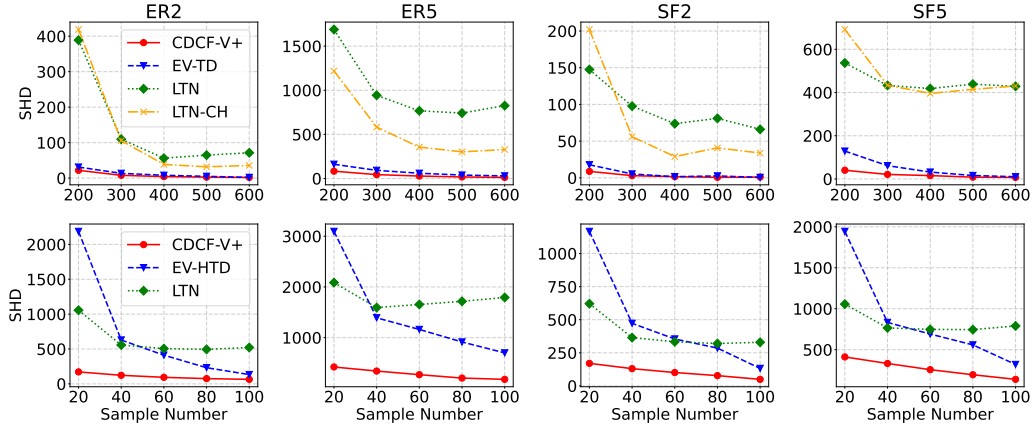

Figure 3.1: Performance (SHD) tested on 100 nodes graph recovering from different sample number.

Table 3.1: Results of 50, 100, 1000 nodes on 3000 linear Gaussian SEM samples.

| Nodes | Graph | NOTEARS | DAG-GNN | CORL-2 | NPVAR | EV-TD | CDCF-V+ |
|---|---|---|---|---|---|---|---|
| 50 | ER2 | $38.6_{10.8}$ | $30.6_{8.3}$ | $17.9_{10.6}$ | $0.4_{0.5}$ | $\mathbf{0.0_{0.0}}$ | $\mathbf{0.0_{0.0}}$ |
| | ER5 | $67.8_{7.5}$ | $93.2_{109.4}$ | $64.8_{13.1}$ | $0.6_{0.8}$ | $0.1_{0.3}$ | $\mathbf{0.0_{0.0}}$ |
| | SF2 | $3.5_{1.6}$ | $79.3_{93.2}$ | $\mathbf{0.0_{0.0}}$ | $1.1_{1.0}$ | $\mathbf{0.0_{0.0}}$ | $\mathbf{0.0_{0.0}}$ |
| | SF5 | $20.1_{14.3}$ | $89.2_{99.2}$ | $20.8_{10.1}$ | $1.0_{0.9}$ | $\mathbf{0.0_{0.0}}$ | $\mathbf{0.0_{0.0}}$ |
| 100 | ER2 | $72.6_{23.5}$ | $66.2_{19.2}$ | $18.6_{5.7}$ | $2.1_{1.2}$ | $\mathbf{0.0_{0.0}}$ | $\mathbf{0.0_{0.0}}$ |
| | ER5 | $170.3_{34.2}$ | $236.4_{36.8}$ | $164.8_{17.1}$ | $2.3_{1.2}$ | $0.2_{0.4}$ | $\mathbf{0.1_{0.3}}$ |
| | SF2 | $2.3_{1.3}$ | $156.8_{21.2}$ | $\mathbf{0.0_{0.0}}$ | $3.0_{1.41}$ | $\mathbf{0.0_{0.0}}$ | $\mathbf{0.0_{0.0}}$ |
| | SF5 | $90.2_{34.5}$ | $165.2_{22.0}$ | $10.8_{6.1}$ | $2.7_{0.9}$ | $0.1_{0.3}$ | $\mathbf{0.0_{0.0}}$ |
| 1000 | ER2 | - | - | - | - | $0.4_{0.5}$ | $\mathbf{0.1_{0.3}}$ |
| | ER5 | - | - | - | - | $21.8_{3.8}$ | $\mathbf{8.9_{4.2}}$ |
| | SF2 | - | - | - | - | $\mathbf{0.0_{0.0}}$ | $\mathbf{0.0_{0.0}}$ |
| | SF5 | - | - | - | - | $0.3_{0.5}$ | $\mathbf{0.0_{0.0}}$ |

Table 3.2: Running time (seconds) on 30 and 100 nodes over 3000 sample.

| | 30 | | | | 100 | | | |
|---|---|---|---|---|---|---|---|---|
| | ER2 | ER5 | SF2 | SF5 | ER2 | ER5 | SF2 | SF5 |
| CDCF-$*$ | **0.004** | **0.005** | **0.004** | **0.005** | **0.017** | **0.016** | **0.016** | **0.017** |
| EV-TD | 0.19 | 0.16 | 0.12 | 0.12 | 14.42 | 12.88 | 15.04 | 14.78 |
| LISTEN | 0.26 | 0.13 | 0.13 | 0.14 | 13.97 | 13.41 | 13.42 | 15.43 |
| EV-HTD | 8.27 | 7.48 | 6.72 | 12.50 | 260.74 | 302.36 | 241.59 | 387.92 |
| DAG-GNN | 49.15 | 49.02 | 38.44 | 41.03 | 137.25 | 238.71 | 158.13 | 187.21 |
| NPVAR | 84.24 | 82.57 | 108.37 | 109.13 | 9867.96 | 9084.78 | 10667.88 | 10173.89 |
| NOTEARS | 78.19 | 597.16 | 51.57 | 306.31 | 3237.8 | 1803.30 | 880.19 | 4159.82 |
| CORL1 | 17573.08 | 18799.21 | 16422.11 | 16588.30 | – | – | – | – |

Table 3.2 shows the running time which is tested on a 2.3 GHz single Intel Core i5 CPU. Besides, parallel calculation of the matrix multiplication on GPU makes the algorithm even faster. Recovering 5000 and 10000 nodes graph from 3000 samples on an A100 Nvidia GPU is approximately 400 and 2400 seconds, respectively. For comparison, EV-TD costs approximately 100 hours to recover a 1000 nodes DAG from 3000 samples. As illustrated in the table, CDCF is approximately dozens or hundreds of times faster than EV-TD and LISTEN, and tens of thousands times faster than NOTEARS as CDCF does not have to update the parameters with gradients.

Table 3.3: Results on Proteins data sets.

| Data sets | Methods | FDR | TPR | FPR | SHD | N | P | F1 |
|---|---|---|---|---|---|---|---|---|
| 853 samples 17 edges | CDCF-V/V+ | 0.533 | **0.412** | 0.210 | 11 | 15 | 0.467 | 0.438 |
| | CDCF-S/S+ | **0.500** | **0.412** | **0.184** | **10** | 14 | **0.500** | **0.452** |
| | CDCF-VS/VS+ | **0.500** | **0.412** | **0.184** | **10** | 14 | **0.500** | **0.452** |
| | NOTEARS | 0.588 | **0.412** | 0.263 | 13 | 17 | 0.412 | 0.412 |
| | NOTEARSMLP | 0.733 | 0.235 | 0.290 | 18 | 15 | 0.267 | 0.250 |
| | CORL1&2 | 0.533 | **0.412** | 0.211 | 11 | 15 | 0.467 | 0.438 |
| | EV-TD | 0.645 | 0.294 | 0.237 | 17 | 14 | 0.357 | 0.323 |
| | LISTEN | 0.750 | 0.176 | 0.237 | 18 | 12 | 0.250 | 0.207 |
| | NPVAR | 0.800 | 0.176 | 0.316 | 19 | 15 | 0.200 | 0.188 |
| | DAG-GNN | 0.588 | **0.412** | 0.263 | 15 | 17 | 0.412 | 0.412 |
| 7466 samples 20 edges | CDCF-V/V+ | 0.667 | 0.400 | 0.457 | 21 | 24 | 0.333 | 0.364 |
| | CDCF-S/S+ | 0.611 | 0.350 | 0.314 | 17 | 18 | 0.389 | 0.368 |
| | CDCF-VS/VS+ | **0.556** | 0.400 | **0.286** | **16** | 18 | **0.444** | **0.421** |
| | NOTEARS | 0.650 | 0.350 | 0.371 | 20 | 20 | 0.350 | 0.350 |
| | NOTEARSMLP | 0.800 | 0.200 | 0.457 | 26 | 20 | 0.200 | 0.200 |
| | CORL1&2 | 0.667 | 0.400 | 0.457 | 21 | 24 | 0.333 | 0.363 |
| | EV-TD | 0.700 | 0.300 | 0.400 | 25 | 20 | 0.300 | 0.300 |
| | LISTEN | 0.714 | 0.300 | 0.429 | 23 | 21 | 0.286 | 0.293 |
| | NPVAR | 0.679 | **0.450** | 0.543 | 24 | 28 | 0.321 | 0.375 |
| | DAG-GNN | 0.650 | 0.350 | 0.371 | 20 | 20 | 0.350 | 0.350 |

Due to the page limitation, further experiments and discussions of the ablation study (Figures B.3 to B.14, Tables B.1 to B.6), choice of $\lambda$ (Tables B.7 to B.10), and performances on different noise distribution (Figures B.1, B.2) and deviation (Tables B.11, B.12, B.13) are given in Appendix B.

## 3.2 PROTEINS DATA SET

We consider a bioinformatics data set (Sachs et al., 2005) consisting of continuous measurements of expression levels of proteins and phospholipids in the human immune system cells. This is a widely used data set for research on graphical models, with experimental annotations accepted by the biological research community. Following the previous algorithms setting, we noticed that different previous papers adopted different observations. To included them all, we considered the observational 853 samples from the "CD3, CD28" simulation tested by Teyssier & Koller (2005); Lachapelle et al. (2020); Zhu et al. (2020) and all 7466 samples from nine different simulations tested by Zheng et al. (2018; 2020); Yu et al. (2019).

We report the experimental results on both settings in Table 3.3. The implementation codes of the baselines are introduced in the appendix, and we use the default settings of the hyper-parameters provided in their codes. The evaluate metric is FDR, TPR, FPR, SHD, predicted nodes number (N), precision (P), F1 score. As the recall score is equal to TPR, we do not include it in the table. In both settings, CDCF-VS+ achieves state-of-the-art performance. [1] Several reasons make the recovered graph not exactly the same as the expected one. The ground truth graph suggested by the paper is mixed with directed and indirect edges. Under the settings of SEM, the node "PKA" is quite similar to the leaf nodes since most of its edges are indirect while the ground truth graph notes it as the out edges. Non-linear would not be an impact issue here since NOTEARS and our algorithm both achieve decent results. In the meantime, we do not deny that further extension of our algorithm to non-linear representation would witness an improvement on this data set.

## 3.3 KNOWLEDGE BASE DATA SET

We test our algorithm on FB15K-237 data set (Toutanova et al., 2015) in which the knowledge is organized as $\{Subject, Predicate, Object\}$ triplets. The data set has 15K triplets and 237 types of predicates. In this experiment, we only consider the single jump predicate between the entities, which

---

[1] For NOTEARS-MLP, Table 3.3 reported the results reproduced by the code provided in Zheng et al. (2020).

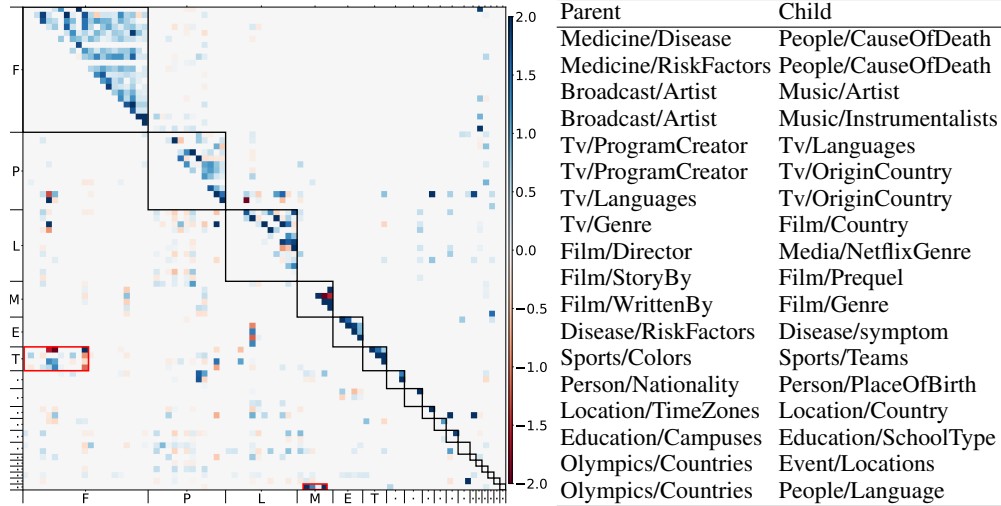

| Parent | Child |
|---|---|
| Medicine/Disease | People/CauseOfDeath |
| Medicine/RiskFactors | People/CauseOfDeath |
| Broadcast/Artist | Music/Artist |
| Broadcast/Artist | Music/Instrumentalists |
| Tv/ProgramCreator | Tv/Languages |
| Tv/ProgramCreator | Tv/OriginCountry |
| Tv/Languages | Tv/OriginCountry |
| Tv/Genre | Film/Country |
| Film/Director | Media/NetflixGenre |
| Film/StoryBy | Film/Prequel |
| Film/WrittenBy | Film/Genre |
| Disease/RiskFactors | Disease/symptom |
| Sports/Colors | Sports/Teams |
| Person/Nationality | Person/PlaceOfBirth |
| Location/TimeZones | Location/Country |
| Education/Campuses | Education/SchoolType |
| Olympics/Countries | Event/Locations |
| Olympics/Countries | People/Language |

Figure 3.2: The recovered weighted adjacent matrix (left) and examples of the high confidence relation pairs (right) on FB15k-237 dataset.

have 97 predicates remained. We want to discover the causal relationships between the predicates. We organize the observation data as each sample corresponds to an entity with awareness of the position (Subject or Object), and each variable corresponds to a predicate in this knowledge base.

In Figure 3.2, we give the adjacent weighted matrix of the generated graph and several examples with high confidence (larger than 0.5). In the left figure, the label of the axis notes the first capital letter of the domain of the relations. Some of them are replaced with a dot to save space. The exact domain name and the picture with the full predicate name are provided in the appendix. The domain clusters are denoted in black boxes at the diagonal of the adjacent matrix. The red boxes denoted the cross-domain relations that are worth paying attention to. Consistent with the innateness of human sense, the recovered relationships inside a domain are denser than those across domains. In the cross-domain relations, we found that the predicate in domain "TV" ("T") has many relations with the domain "Film" ("F"), the domain "Broadcast" (last row) have many relations with the domain "Music" ("M"). Several cases of the predicted causal relationships are listed on the right side of Figure 3.2, we can see that the discovered indication relations between predicates are quite reasonable.

## 4    CONCLUSION AND FUTURE WORK

In this paper, we proposed a topology search algorithm for the DAG structure recovery problem. Our algorithm is better than the existing methods in both time and sample complexities. To be specific, the time complexity of our algorithm is $\mathcal{O}(p^2 n + p^3)$, while the fastest algorithm before is $\mathcal{O}(p^5 n)$ (Park, 2020; Gao et al., 2020), where $p$ and $n$ are the numbers of nodes and samples, respectively. Under different assumptions, our algorithm takes $\mathcal{O}(\log(p))$ or $\mathcal{O}(p)$ samples to exactly recover the DAG structure. Experimental results on synthetic data sets, proteins data sets, and knowledge base data set demonstrate the efficiency and effectiveness of our algorithm. For synthetic data sets, compared with previous baselines, our algorithm improves the performance with a significant margin and at least tens or hundreds of times faster. For the proteins data set, we achieve state-of-the-art performance. For the knowledge base data set, we can observe many reasonable structures of the discovered DAG.

The proposed algorithm is under the assumption of linear SEM. Generalization of CDCF to nonlinear SEM would be a valuable and important research topic. Learning the representation of the observed data for better structure reconstruction via the CDCF algorithm, which requires the algorithm differentiable, is also an attractive problem. To deal with the extremely large-scale problems, such as millions of nodes, implementing CDCF via sparse matrix storage and calculation on the GPU is a promising way to further improve computational performance.

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

## A  PROOF OF THEOREM 2.1

In this section, we first give several lemmas, then prove Theorem 2.1.

**Lemma A.1** *Let* $\mathbf{x} \in \mathbb{R}^p$ *be a zero-mean random vector,* $\boldsymbol{C} = \mathbb{E}(\mathbf{x}\mathbf{x}^{\mathrm{T}}) \in \mathbb{R}^{p \times p}$ *be the covariance matrix. Let* $\mathbf{x}_1, \ldots, \mathbf{x}_n$ *be* $n$ *independent samples,* $\widehat{\boldsymbol{C}} = \frac{1}{n} \sum_{k=1}^{n} \mathbf{x}_k \mathbf{x}_k^{\mathrm{T}}$ *be the sample covariance estimator. Assume* $\|\boldsymbol{C} - \widehat{\boldsymbol{C}}\| \leq \epsilon$ *for some* $\epsilon > 0$. *Denote* $\widehat{\boldsymbol{C}}_\lambda = \widehat{\boldsymbol{C}} + \lambda \boldsymbol{I}$, *where* $\lambda = \mathcal{O}(\epsilon) \geq 0$ *is a parameter. Let the Cholesky factorizations of* $\boldsymbol{C} = \mathbb{E}\mathbf{x}\mathbf{x}^{\mathrm{T}}$ *and* $\widehat{\boldsymbol{C}}_\lambda$ *be* $\boldsymbol{C} = \boldsymbol{L}\boldsymbol{L}^{\mathrm{T}}$ *and* $\widehat{\boldsymbol{C}}_\lambda = \widehat{\boldsymbol{L}}\widehat{\boldsymbol{L}}^{\mathrm{T}}$, *respectively, where* $\boldsymbol{L}$ *and* $\widehat{\boldsymbol{L}}$ *are both lower triangular. If* $\|\boldsymbol{L}^{-1}\|^2(\epsilon + \lambda) < \frac{3}{4}$, *then*

$$\left|\|\boldsymbol{L}_{i,:}\|^2 - \|[\widehat{\boldsymbol{L}}_\lambda]_{i,:}\|^2\right| \leq \epsilon + \lambda = \mathcal{O}(\epsilon), \qquad\qquad \text{for } 1 \leq i \leq p; \qquad (9)$$

$$\left|[\boldsymbol{L}^{-1}]_{ij} - [\widehat{\boldsymbol{L}}^{-1}]_{ij}\right| \leq 4\|\boldsymbol{L}^{-1}\|_{2,\infty}^2\|\boldsymbol{L}^{-\mathrm{T}}\|_{2,\infty}(\epsilon + \lambda) = \mathcal{O}(\epsilon), \qquad \text{for } i > j. \qquad (10)$$

*Proof.*  For all $1 \leq i \leq p$, we have

$$\left|\|\boldsymbol{L}_{i,:}\|^2 - \|\widehat{\boldsymbol{L}}_{i,:}\|^2\right| = |\boldsymbol{C}_{ii} - [\widehat{\boldsymbol{C}}_\lambda]_{ii}| \leq \|\boldsymbol{C} - \widehat{\boldsymbol{C}}_\lambda\| \leq \|\boldsymbol{C} - \widehat{\boldsymbol{C}}\| + \lambda \leq \epsilon + \lambda, \qquad (11)$$

which completes the proof for (9).

Next, we show (10). Let

$$\boldsymbol{L}^{-1}\widehat{\boldsymbol{L}} = \boldsymbol{I} + \boldsymbol{F}, \qquad (\boldsymbol{I} + \boldsymbol{F})(\boldsymbol{I} + \boldsymbol{F})^{\mathrm{T}} = \boldsymbol{I} + \boldsymbol{E}. \qquad (12)$$

We know that

$$\widehat{\boldsymbol{L}}^{-1} - \boldsymbol{L}^{-1} = [(\boldsymbol{I} + \boldsymbol{F})^{-1} - \boldsymbol{I}]\boldsymbol{L}^{-1} = -\boldsymbol{F}(\boldsymbol{I} + \boldsymbol{F})^{-1}\boldsymbol{L}^{-1}, \qquad (13)$$

$$\boldsymbol{E} = \boldsymbol{L}^{-1}\widehat{\boldsymbol{L}}\widehat{\boldsymbol{L}}^{\mathrm{T}}\boldsymbol{L}^{-\mathrm{T}} - \boldsymbol{I} = \boldsymbol{L}^{-1}(\widehat{\boldsymbol{C}}_\lambda - \boldsymbol{C})\boldsymbol{L}^{-\mathrm{T}}. \qquad (14)$$

Then it follows from (13) that for $i > j$

$$\left|[\boldsymbol{L}^{-1}]_{ij} - [\widehat{\boldsymbol{L}}^{-1}]_{ij}\right| \leq \|\boldsymbol{F}_{i,1:i-1}\|\|[(\boldsymbol{I}+\boldsymbol{F})^{-1}\boldsymbol{L}^{-1}]_{:,j}\| \leq \|\boldsymbol{F}_{i,1:i-1}\|\|(\boldsymbol{I}+\boldsymbol{F})^{-1}\|\|\boldsymbol{L}^{-\mathrm{T}}\|_{2,\infty}. \quad (15)$$

First, we give an upper bound for $\|(\boldsymbol{I} + \boldsymbol{F})^{-1}\|$. Using (12), we have $(\boldsymbol{I} + \boldsymbol{F})^{-\mathrm{T}}(\boldsymbol{I} + \boldsymbol{F})^{-1} = (\boldsymbol{I} + \boldsymbol{E})^{-1}$. It follows

$$\|(\boldsymbol{I} + \boldsymbol{F})^{-1}\| = \|(\boldsymbol{I} + \boldsymbol{F})^{-\mathrm{T}}(\boldsymbol{I} + \boldsymbol{F})^{-1}\|^{\frac{1}{2}} = \|(\boldsymbol{I} + \boldsymbol{E})^{-1}\|^{\frac{1}{2}}$$
$$\leq \frac{1}{\sqrt{1 - \|\boldsymbol{E}\|}} \leq \frac{1}{\sqrt{1 - \|\boldsymbol{L}^{-1}\|^2\|\widehat{\boldsymbol{C}}_\lambda - \boldsymbol{C}\|}}, \qquad (16)$$

where the last inequality uses (14).

Second, we give upper bound for $\|\boldsymbol{F}_{i,1:i-1}\|$. It follows from the second equality of (12) that

$$(1 + \boldsymbol{F}_{ii})^2 + \|\boldsymbol{F}_{i,1:i-1}\|^2 = 1 + \boldsymbol{E}_{ii}. \qquad (17)$$

Therefore,

$$\|\boldsymbol{F}_{i,1:i-1}\|^2 \leq |(1 + \boldsymbol{F}_{ii})^2 - 1| + \boldsymbol{E}_{ii} \overset{(a)}{\leq} \frac{\widehat{\boldsymbol{L}}_{ii}^2 - \boldsymbol{L}_{ii}^2}{\boldsymbol{L}_{ii}^2} + \boldsymbol{E}_{ii}$$
$$\overset{(b)}{\leq} \frac{\epsilon + \lambda}{\boldsymbol{L}_{ii}^2} + \|\boldsymbol{L}^{-1}\|_{2,\infty}^2\|\widehat{\boldsymbol{C}}_\lambda - \boldsymbol{C}\| \overset{(c)}{\leq} 2\|\boldsymbol{L}^{-1}\|_{2,\infty}^2(\epsilon + \lambda), \qquad (18)$$

where (a) uses (12), (b) uses (9) and (14), (c) uses $\|\boldsymbol{C} - \widehat{\boldsymbol{C}}\| \leq \epsilon$. Substituting (18) and (16) into (15), we get

$$\left|[\boldsymbol{L}^{-1}]_{ij} - [\widehat{\boldsymbol{L}}^{-1}]_{ij}\right| \leq 2\|\boldsymbol{L}^{-1}\|_{2,\infty}^2\|\boldsymbol{L}^{-\mathrm{T}}\|_{2,\infty} \frac{\epsilon + \lambda}{\sqrt{1 - \|\boldsymbol{L}^{-1}\|^2(\epsilon + \lambda)}}. \qquad (19)$$

The conclusion follows since $\|\boldsymbol{L}^{-1}\|^2(\epsilon + \lambda) < \frac{3}{4}$.  $\qquad\qquad\square$

**Theorem A.2** *Let $\mathbf{x} \in \mathbb{R}^p$ be a zero-mean random vector, $\boldsymbol{C} = \mathbb{E}(\mathbf{x}\mathbf{x}^{\mathrm{T}}) \in \mathbb{R}^{p \times p}$ be the covariance matrix. Let $\mathbf{x}_1, \ldots, \mathbf{x}_n$ be $n$ independent samples, $\widehat{\boldsymbol{C}} = \frac{1}{n}\sum_{k=1}^{n} \mathbf{x}_k \mathbf{x}_k^{\mathrm{T}}$ be the sample covariance estimator. Assume $\|\boldsymbol{C} - \widehat{\boldsymbol{C}}\| \le \epsilon$ for some $\epsilon > 0$. Denote $\widehat{\boldsymbol{C}}_\lambda = \widehat{\boldsymbol{C}} + \lambda \boldsymbol{I}$, where $\lambda = \mathcal{O}(\epsilon) \ge 0$ is a parameter. Let the Cholesky factorizations of $\boldsymbol{C} = \mathbb{E}\mathbf{x}\mathbf{x}^{\mathrm{T}}$ and $\widehat{\boldsymbol{C}}_\lambda$ be $\boldsymbol{C} = \boldsymbol{L}\boldsymbol{L}^{\mathrm{T}}$ and $\widehat{\boldsymbol{C}}_\lambda = \widehat{\boldsymbol{L}}\widehat{\boldsymbol{L}}^{\mathrm{T}}$, respectively, where $\boldsymbol{L}$ and $\widehat{\boldsymbol{L}}$ are both lower triangular. For the linear SEM model* (1), *assume* (2) *and* (4), *and for $k \in Pa_{\mathcal{G}}(j)$, $\delta = \inf_{k \in Pa_{\mathcal{G}}(j)} \delta_{jk} > 0$, where*

$$\delta_{jk} = \sigma_{i_j^*}^2 + \|\widehat{\boldsymbol{\Sigma}}_n[(\boldsymbol{I} - \boldsymbol{T})^{-1}]_{k:j-1,k}\|^2 - \sigma_{i_k^*}^2.$$

*If $\delta \ge 4(\epsilon + \lambda)$ and $\|\boldsymbol{L}^{-1}\|^2(\epsilon + \lambda) < \frac{3}{4}$, then* CDCF-V *is able to recover $\boldsymbol{P}$ exactly. In addition, it holds that*

$$\|\mathrm{TRIU}(\boldsymbol{U}_p) - \boldsymbol{T}\|_{\max} \le 4\|\widehat{\boldsymbol{\Sigma}}_*^{-1}(\boldsymbol{I} - \boldsymbol{T})^{\mathrm{T}}\|_{2,\infty}^2 \|(\boldsymbol{I} - \boldsymbol{T})\widehat{\boldsymbol{\Sigma}}_*^{-\mathrm{T}}\|_{2,\infty}(\epsilon + \lambda),$$

*where* $\mathrm{TRIU}(\boldsymbol{U}_p)$ *stands for the strictly upper triangular part of $\boldsymbol{U}_p$, $\boldsymbol{U}_p$ is the output of outer loop of Algorithm* 1 *with criterion* (V).

*Proof.* For SEM model (1), denote $\widehat{\boldsymbol{C}}_* = \mathbb{E}(\frac{1}{n}\widehat{\boldsymbol{X}}^{\mathrm{T}}\widehat{\boldsymbol{X}})$, $\widehat{\boldsymbol{\Sigma}}_*^2 = \mathbb{E}(\frac{1}{n}\widehat{\boldsymbol{N}}^{\mathrm{T}}\widehat{\boldsymbol{N}}) = \widehat{\boldsymbol{\Sigma}}_n^{\mathrm{T}}\widehat{\boldsymbol{\Sigma}}_n$, we have (5), i.e.,

$$\widehat{\boldsymbol{C}}_* = (\boldsymbol{I} - \boldsymbol{T})^{-\mathrm{T}}\widehat{\boldsymbol{\Sigma}}_*^2(\boldsymbol{I} - \boldsymbol{T})^{-1} = (\boldsymbol{I} - \boldsymbol{T})^{-\mathrm{T}}\widehat{\boldsymbol{\Sigma}}_n^{\mathrm{T}}\widehat{\boldsymbol{\Sigma}}_n(\boldsymbol{I} - \boldsymbol{T})^{-1}. \tag{20}$$

When the permutation $\boldsymbol{i}^* = [i_1^*, \ldots, i_p^*]$ is exactly recovered, then $\boldsymbol{U}_p$ in CDCF-V satisfies

$$\widehat{\boldsymbol{C}}_\lambda = \frac{1}{n}\boldsymbol{X}_{:,\boldsymbol{i}^*}^{\mathrm{T}}\boldsymbol{X}_{:,\boldsymbol{i}^*} + \lambda \boldsymbol{I} = \boldsymbol{U}_p^{-\mathrm{T}}\boldsymbol{U}_p^{-1}. \tag{21}$$

Denote $\boldsymbol{i}_j^* = [i_1^*, \ldots, i_j^*]$ for all $j = 1, \ldots, p$. Consider the $k$th diagonal entries of (20) and (21). By calculations, we get

$$[\widehat{\boldsymbol{C}}_*]_{kk} = [(\boldsymbol{I} - \boldsymbol{T})^{-1}]_{:,k}^{\mathrm{T}}\widehat{\boldsymbol{\Sigma}}_n^{\mathrm{T}}\widehat{\boldsymbol{\Sigma}}_n[(\boldsymbol{I} - \boldsymbol{T})^{-1}]_{:,k} = \sigma_{i_k^*}^2 + \|\boldsymbol{u}_k\|^2, \tag{22}$$

$$[\widehat{\boldsymbol{C}}_\lambda]_{kk} = \frac{1}{n}\|\boldsymbol{X}_{i_k^*}\|^2 + \lambda = \frac{1}{\boldsymbol{U}_{kk}^2} + \|\widehat{\boldsymbol{u}}_k\|^2, \tag{23}$$

where

$$\boldsymbol{u}_k = [\widehat{\boldsymbol{\Sigma}}_n]_{1:k-1,1:k-1}(\boldsymbol{I}_{k-1} - \boldsymbol{T}_{1:k-1,1:k-1})^{-1}\boldsymbol{T}_{1:k-1,k}, \quad \widehat{\boldsymbol{u}}_k = \frac{1}{n}\boldsymbol{U}_{k-1}^{\mathrm{T}}\boldsymbol{X}_{:,\boldsymbol{i}_{k-1}^*}^{\mathrm{T}}\boldsymbol{X}_{:,i_k^*}. \tag{24}$$

Using $\|\boldsymbol{C} - \widehat{\boldsymbol{C}}\| \le \epsilon$, we have

$$|[\widehat{\boldsymbol{C}}_*]_{kk} - [\widehat{\boldsymbol{C}}_\lambda]_{kk}| \le \|\boldsymbol{C} - \widehat{\boldsymbol{C}}_\lambda\| \le \|\boldsymbol{C} - \widehat{\boldsymbol{C}}\| + \lambda \le \epsilon + \lambda. \tag{25}$$

By Lemma A.1, we have

$$|\|\boldsymbol{u}_k\|^2 - \|\widehat{\boldsymbol{u}}_k\|^2| \le \epsilon + \lambda. \tag{26}$$

Using (22), (23), (25) and (26), we get

$$|\sigma_{i_k^*}^2 - \frac{1}{\boldsymbol{U}_{kk}^2}| \le 2(\epsilon + \lambda). \tag{27}$$

Assume that $i_1^*, \ldots, i_{k-1}^*$ $(k \ge 1)$ are all correctly recovered. And without loss of generality, for $k \in Pa_{\mathcal{G}}(j)$, we also assume $\boldsymbol{T}_{k:j-1,j} \ne 0$ (otherwise, $j$th and $k$th columns are exchangeable, and $\boldsymbol{i}$ forms another equivalence topology order to the same DAG (Sedgewick & Wayne, 2011)). Then we

have for $k \in Pa_{\mathcal{G}}(j)$ that

$$
\begin{aligned}
\frac{1}{n}\|\boldsymbol{X}_{i_j^*}\|^2 + \lambda - \|[\widehat{\boldsymbol{u}}_j]_{1:k-1}\|^2 & \overset{(a)}{=} [\widehat{\boldsymbol{C}}_*]_{jj} + [\widehat{\boldsymbol{C}}_\lambda]_{jj} - [\widehat{\boldsymbol{C}}_*]_{jj} - \|[\widehat{\boldsymbol{u}}_j]_{1:k-1}\|^2 \\
& \overset{(b)}{\geq} [\widehat{\boldsymbol{C}}_*]_{jj} - (\epsilon + \lambda) - \|[\boldsymbol{u}_j]_{1:k-1}\|^2 - (\epsilon + \lambda) \\
& \overset{(c)}{=} \sigma_{i_j^*} + \|[\boldsymbol{u}_j]_{k:j-1}\|^2 - 2(\epsilon + \lambda) \\
& \overset{(d)}{\geq} \sigma_{i_k^*} + \delta - 2(\epsilon + \lambda) \\
& \overset{(e)}{=} [\widehat{\boldsymbol{C}}_*]_{kk} - \|\boldsymbol{u}_k\|^2 + \delta - 2(\epsilon + \lambda) \\
& \overset{(f)}{\geq} [\widehat{\boldsymbol{C}}_\lambda]_{kk} - \|\widehat{\boldsymbol{u}}_k\|^2 + \delta - 4(\epsilon + \lambda) \\
& \overset{(g)}{=} \frac{1}{n}\|\boldsymbol{X}_{i_k^*}\|^2 + \lambda - \|\widehat{\boldsymbol{u}}_k\|^2 + \delta - 4(\epsilon + \lambda),
\end{aligned}
$$

where (a) uses (23), (b) and (f) uses (25) and Lemma A.1, (c) uses (22), (d) dues to the assumption $\sigma_{i_j^*} \geq \sigma_{i_k^*}$ for $k \in Pa_{\mathcal{G}}(j)$, (e) uses (22), (g) uses (23). Therefore, using $\delta > 4(\epsilon + \lambda)$, we have

$$
\frac{1}{n}\|\boldsymbol{X}_{i_j^*}\|^2 + \lambda - \|[\widehat{\boldsymbol{u}}_j]_{1:k-1}\|^2 > \frac{1}{n}\|\boldsymbol{X}_{i_k^*}\|^2 + \lambda - \|\widehat{\boldsymbol{u}}_k\|^2,
$$

which implies that $i_k^*$ can be correctly recovered. So, overall speaking, CDCF-V is able to recover the permutation $\boldsymbol{P}$.

The upper bound for $\|\mathrm{TRIU}(\boldsymbol{U}_p) - \boldsymbol{T}\|_{\max}$ follows from Lemma A.1. The proof is completed. $\quad\square$

**Proposition 2** *Let $\boldsymbol{N}_{i,:}$ be independent bounded, or sub-Gaussian,* [2] *or regular polynomial-tail,* [3] *then for $n > N(\epsilon)$, it holds $\|\widehat{\boldsymbol{C}}_{xx} - \boldsymbol{C}_{xx}\| \leq \epsilon$, w.h.p. Specifically,*

$$
\begin{aligned}
& N(\epsilon) \geq C_1 \ \log p \ \left( \frac{\|(\boldsymbol{I} - \boldsymbol{T})^{-1}\|^2 \|\boldsymbol{C}_{nn}\|}{\epsilon} \right)^2, && \text{for bounded class;} \\
& N(\epsilon) \geq C_2 \ p \ \left( \frac{\|(\boldsymbol{I} - \boldsymbol{T})^{-1}\|^2 \|\boldsymbol{C}_{nn}\|}{\epsilon} \right)^2, && \text{for the sub-Gaussian class;} \\
& N(\epsilon) \geq C_3 \ p \ \left( \frac{\|(\boldsymbol{I} - \boldsymbol{T})^{-1}\|^2 \|\boldsymbol{C}_{nn}\|}{\epsilon} \right)^{2(1+r^{-1})}, && \text{for the regular polynomial tail class.}
\end{aligned}
$$

*Proof.* For SEM model (1), we have

$$
\|\widehat{\boldsymbol{C}}_{xx} - \boldsymbol{C}_{xx}\| \leq \|(\boldsymbol{I} - \boldsymbol{T})^{-1}\|^2 \|\widehat{\boldsymbol{C}}_{nn} - \boldsymbol{C}_{nn}\| \leq \|(\boldsymbol{I} - \boldsymbol{T})^{-1}\|^2 \|\boldsymbol{C}_{nn}\| \|\boldsymbol{C}_{nn}^{-\frac{1}{2}} \widehat{\boldsymbol{C}}_{nn} \boldsymbol{C}_{nn}^{-\frac{1}{2}} - \boldsymbol{I}\|, \quad (28)
$$

where $\boldsymbol{C}_{xx} = \mathbb{E}\mathbf{x}\mathbf{x}^{\mathrm{T}}$, $\boldsymbol{C}_{nn} = \mathbb{E}\mathbf{n}\mathbf{n}^{\mathrm{T}}$ are the covariance matrices for $\mathbf{x}$ and $\mathbf{n}$, respectively, $\widehat{\boldsymbol{C}}_{xx}$, $\widehat{\boldsymbol{C}}_{nn}$ are the sample covariance matrices for $\mathbf{x}$ and $\mathbf{n}$, respectively. The three results listed above follow from Corollary 5.52, Theorem 5.39 in Vershynin (2010), Theorem 1.1 in Srivastava & Vershynin (2013), respectively. $\quad\square$

---

[2] A random vector $\mathbf{z}$ is isotropic and sub-Gaussian if $\mathbb{E}\mathbf{z}\mathbf{z}^{\mathrm{T}} = \boldsymbol{I}$ and and there exists constant $C > 0$ such that $\mathbb{P}(|\boldsymbol{v}^{\mathrm{T}}\mathbf{z}| > t) \leq \exp(-Ct^2)$ for any unit vector $\boldsymbol{v}$. Here by "$\boldsymbol{N}_{i,:}$ is sub-Gaussian" we mean that $\boldsymbol{C}_{nn}^{-\frac{1}{2}} \boldsymbol{N}_{i,:}^{\mathrm{T}}$ is an isotropic and sub-Gaussian random vector.

[3] A random vector $\mathbf{z}$ is isotropic and regular polynomial-tail if $\mathbb{E}\mathbf{z}\mathbf{z}^{\mathrm{T}} = \boldsymbol{I}$ and there exist constants $r > 1$, $C > 0$ such that $\mathbb{P}(\|\boldsymbol{V}\mathbf{z}\|^2 > t) \leq Ct^{-1-r}$ for any orthogonal projection $\boldsymbol{V}$ and any $t > C \cdot \mathrm{rank}(\boldsymbol{V})$. Here by "$\boldsymbol{N}_{i,:}$ is regular polynomial-tail" we mean that $\boldsymbol{C}_{nn}^{-\frac{1}{2}} \boldsymbol{N}_{i,:}^{\mathrm{T}}$ is an isotropic and regular polynomial-tail random vector.

# B   ADDITIONAL EXPERIMENTS

Here we provide implementation details and additional experiment results.

Figures B.1, B.2 provide the results of Gumbel and Exponential noises, respectively. As we can see from the result, our algorithm still performs better than Eqvar method in different noise types.

Tables B.1, B.2 , B.3, B.4, B.5, B.6 give results on 100 nodes over different sample sizes and variances of our CDCF methods. As noted in Algorithm 1, we have V, S, VS as different criteria to select the current column, "+" representing the sample covariance matrix augmented with the scalar matrix $\frac{\log p}{n}\boldsymbol{I}$. The truncation threshold on column $i$ is $\omega_i = 3.5/\alpha_i$, where $\alpha_i$ is the diagonal value of the Cholesky factor. According to the results, the algorithm "V+" achieves the best performance as the sample size is relatively large. When the sample size is small, the criterion according to sparsity shows very effective performance improvement. We also test different choices over $\lambda = \beta\frac{\log p}{n}, \beta \in \{0.0, 1.0..., 9.0\}$, the result is given in Table B.7, B.8, B.9, B.10. Empirically, $\beta \in \{1.0, 2.0\}$ achieves better results. In practice, one can sample a relatively small and labeled sub-graph of the DAG to test the hyper-parameter setting then apply to large unlabeled the DAG graph.

To test the performance limitation of our methods, we provide the results of SHD on different sample number and node number in Figures B.3 to B.14 where the x-axis represents the sample number (in thousand), the y-axis denotes the node number, the color represents the value of $\log_2(\text{SHD} + 1)$ (the brighter the better). We provide the figures for CDCF-V+, CDCF-S+, and CDCF-VS+ on variances graph and noise types. The figures are drawn on the mean results over ten random seeds. The figures show that the graph can be exactly recovered on 800 nodes at approximately 6000 samples. Comparing CDCF-V+ with CDCF-S+, we find that criterion ($\boldsymbol{S}$) damages the performance when the sample number is relatively large. When sample number $\in \{1500, 3000\}$ and node number $\in \{400, 800\}$, CDCF-S+ achieves better performance. Such trend can also be demonstrated in Tables B.1, B.2, B.3. CDCF-VS+ alleviates the poor performance of CDCF-S when the data is sufficient and achieves good performance on real-world data set.

We also test the performance on linear SEM with monotonously increased noise variance. Concretely, assume the topology order is $\boldsymbol{i} = \{i_1, ..., i_p\}$, we set the noise variance of node $k$ as $\sigma_k = 1 + \boldsymbol{i}_k/p$. We test the results on Gaussian, Gumbel, and Exponential noise with monotonous noise variance. The results are reported in Tables B.11, B.12 and B.13. As the results indicated, even with different noise levels, our algorithms achieve good performance and are able to exactly recover the DAG structure when the data is sufficient.

In the result for knowledge base data set, the axis labels of Figure 3.2 are 'Film', 'People', 'Location', 'Music', 'Education', 'Tv', 'Medicine', 'Sports', 'Olympics', 'Award', 'Time', 'Organization', 'Language', 'MediaCommon', 'Influence', 'Dataworld', 'Business', 'Broadcast' from left to right for x-axis and top to bottom for y-axis, respectively. The adjacent matrix plotted here is re-permuted to make the relations in the same domain close to each other. We keep the adjacent matrix inside a domain an upper triangular matrix. Such typology is equivalent to the generated matrix with the original order.

**Baseline Implementations**   The baselines are implemented via the codes provided from the following links:

- NOTEARS, NOTEARS-MLP: https://github.com/xunzheng/notears
- NPVAR: https://github.com/MingGao97/NPVAR
- EQVAR, LISTEN: https://github.com/WY-Chen/EqVarDAG
- CORL: https://github.com/huawei-noah/trustworthyAI/tree/master/gcastle
- DAG-GNN: https://github.com/fishmoon1234/DAG-GNN

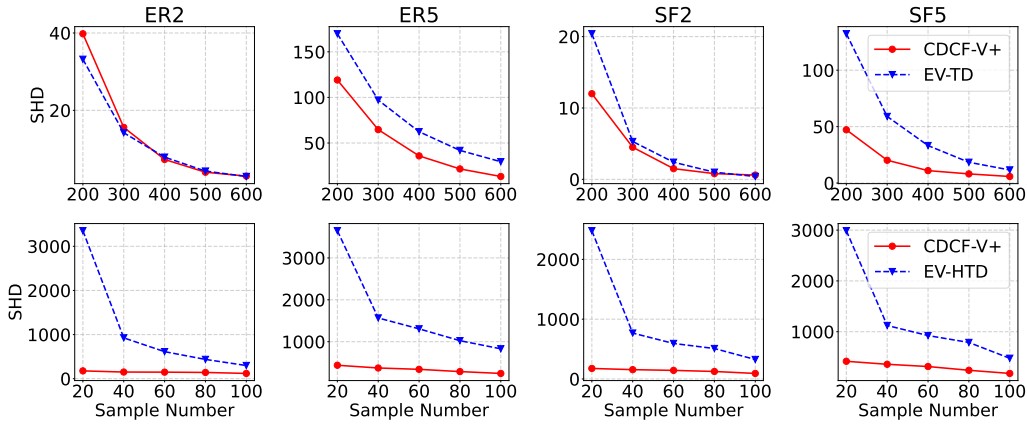

Figure B.1: Performance (SHD) tested on 100 nodes graph recovered from different sample numbers with gumbel noise.

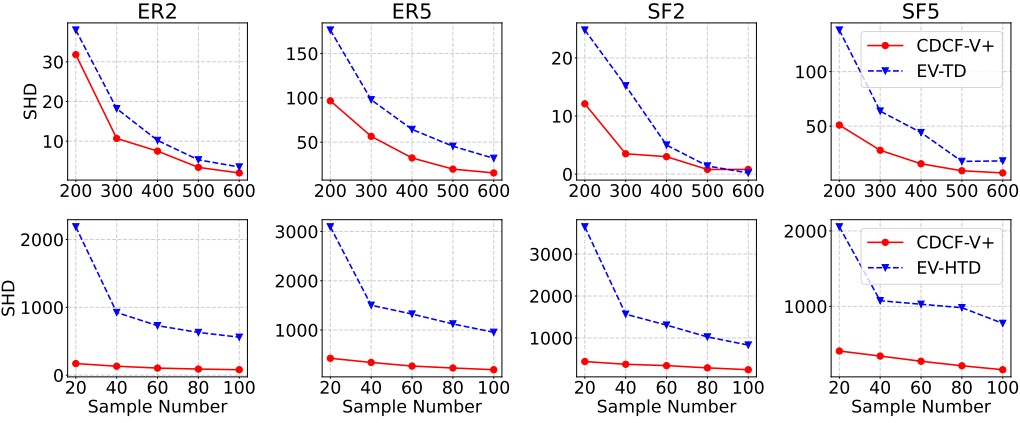

Figure B.2: Performance (SHD) tested on 100 nodes graph recovered from different sample numbers with exponential noise.

Table B.1: SHD results on 100 nodes linear Gaussian noise SEM with low sample size.

|     |     | 20 | 40 | 60 | 80 | 100 |
|-----|-----|------|-------|------|-------|-------|
| ER2 | V+  | 174.4 | **119.3** | **88.7** | **68.7** | **55.7** |
|     | S+  | **173.4** | 120.4 | 92.2 | 71.2 | 57.4 |
|     | VS+ | 174.7 | 126.6 | 99.8 | 75.1 | 70.7 |
| SF2 | V+  | 170.2 | **128.2** | **99.5** | **77.4** | 53.0 |
|     | S+  | 170.1 | 129.2 | 99.8 | 79.5 | **52.1** |
|     | VS+ | **168.8** | 133.6 | 103.6 | 83.6 | 52.9 |
| ER5 | V+  | **427.0** | 356.9 | 284.2 | **223.6** | 194.4 |
|     | S+  | 431.8 | **351.0** | 279.8 | 227.3 | 200.1 |
|     | VS+ | 434.5 | 354.1 | **277.7** | 225.9 | **193.5** |
| SF5 | V+  | 412.6 | **330.0** | 269.2 | 218.2 | 169.5 |
|     | S+  | 409.8 | 332.9 | **266.8** | **215.3** | **167.1** |
|     | VS+ | **409.5** | 337.7 | 270.9 | 220.5 | 173.5 |

Table B.2: SHD results on 100 nodes linear Gumbel noise SEM with low sample size.

| | | 20 | 40 | 60 | 80 | 100 |
|---|---|---|---|---|---|---|
| ER2 | V+ | 177.2 | 151.0 | 148.4 | 141.7 | 119.3 |
| | S+ | **174.8** | **140.5** | **130.6** | **130.2** | **111.7** |
| | VS+ | 229.0 | 256.2 | 293.5 | 258.2 | 223.3 |
| SF2 | V+ | 175.8 | 155.7 | 143.4 | 125.2 | 92.4 |
| | S+ | **171.8** | **144.0** | **125.8** | **102.7** | **78.9** |
| | VS+ | 202.2 | 208.7 | 187.1 | 152.5 | 100.3 |
| ER5 | V+ | 437.3 | 372.1 | 340.1 | 286.3 | **242.0** |
| | S+ | **435.8** | **358.6** | **327.0** | **278.7** | 248.4 |
| | VS+ | 472.1 | 450.1 | 409.5 | 323.5 | 308.2 |
| SF5 | V+ | 417.5 | 358.3 | 314.1 | 240.3 | 178.1 |
| | S+ | **414.2** | **349.8** | **292.0** | **217.8** | **160.0** |
| | VS+ | 443.9 | 424.0 | 342.9 | 258.2 | 181.0 |

Table B.3: SHD results on 100 nodes linear Exponential noise SEM with low sample size.

| | | 20 | 40 | 60 | 80 | 100 |
|---|---|---|---|---|---|---|
| ER2 | V+ | **171.1** | 132.1 | 104.8 | 90.2 | 80.5 |
| | S+ | 171.2 | **128.3** | **95.5** | **83.0** | **71.1** |
| | VS+ | 185.7 | 180.5 | 178.3 | 137.0 | 113.6 |
| SF2 | V+ | **167.0** | 135.1 | 115.4 | 87.4 | 64.8 |
| | S+ | 168.4 | **130.2** | **107.5** | **79.1** | **60.1** |
| | VS+ | 170.7 | 165.0 | 131.3 | 100.4 | 67.0 |
| ER5 | V+ | **427.4** | 342.3 | 268.9 | 229.8 | 194.9 |
| | S+ | 432.5 | **335.1** | **265.8** | **217.3** | **184.4** |
| | VS+ | 454.0 | 353.3 | 282.2 | 238.3 | 213.5 |
| SF5 | V+ | **408.3** | 341.6 | 272.4 | 212.6 | 160.3 |
| | S+ | 408.8 | **336.7** | **265.9** | **204.5** | **152.2** |
| | VS+ | 411.1 | 363.0 | 281.2 | 219.1 | 158.4 |

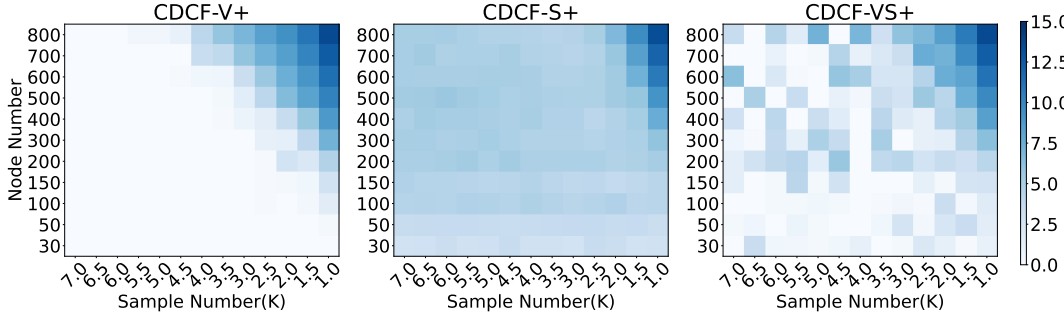

Figure B.3: Performances (log(SHD + 1)) of $n$ as x-axis vs $p$ as y-axis of CDCF-∗+ on ER2 Gaussian noise.

Table B.4: SHD results on 100 nodes linear Gaussian SEM with different sample size.

| | | 200 | 300 | 400 | 500 | 600 | 700 | 800 | 900 | 1000 | 1500 | 2000 | 2500 | 3000 |
|---|---|---|---|---|---|---|---|---|---|---|---|---|---|---|
| ER2 | V | 114.3 | 32.2 | 11.7 | 4.6 | 2.5 | 1.4 | **1.1** | **0.5** | 0.4 | **0.3** | **0.0** | **0.0** | **0.0** |
| | S | 98.4 | 30.3 | 18.6 | 18.4 | 15.3 | 18.5 | 18.2 | 16.0 | 15.4 | 16.5 | 18.1 | 21.9 | 18.5 |
| | VS | 141.6 | 38.7 | 32.3 | 6.1 | 8.0 | 12.5 | 2.5 | 8.6 | 0.6 | **0.3** | 1.2 | **0.0** | 1.0 |
| | V+ | **21.9** | **7.9** | **4.3** | **2.8** | **2.0** | **1.3** | 1.4 | **0.5** | 0.4 | **0.3** | **0.0** | **0.0** | **0.0** |
| | S+ | 29.1 | 17.6 | 13.5 | 17.2 | 14.6 | 16.3 | 18.5 | 16.3 | 15.3 | 16.5 | 18.1 | 21.8 | 18.5 |
| | VS+ | 38.6 | 16.0 | 21.3 | 3.0 | 4.3 | 10.4 | 2.1 | 8.1 | **0.4** | **0.3** | 0.8 | **0.0** | 0.9 |
| SF2 | V | 37.1 | 7.2 | 1.9 | 2.6 | **0.9** | **0.3** | **0.1** | **0.1** | 0.2 | **0.0** | **0.0** | **0.0** | **0.0** |
| | S | 27.3 | 7.5 | 6.9 | 3.8 | 4.2 | 4.9 | 3.9 | 3.3 | 3.6 | 5.3 | 5.7 | 3.4 | 4.8 |
| | VS | 53.7 | 15.4 | 5.0 | 2.4 | 1.9 | 1.3 | 1.1 | 0.2 | 0.2 | 0.1 | **0.0** | 0.3 | **0.0** |
| | V+ | **8.8** | **3.0** | **1.6** | 0.7 | 1.1 | 0.5 | 0.2 | 0.2 | **0.1** | **0.0** | **0.0** | **0.0** | **0.0** |
| | S+ | 11.7 | 4.8 | 7.3 | 3.6 | 4.8 | 5.0 | 4.0 | 3.3 | 4.4 | 5.3 | 5.7 | 3.4 | 4.8 |
| | VS+ | 10.8 | 4.1 | 2.1 | **0.6** | 1.3 | 0.4 | 1.6 | 0.2 | **0.1** | 0.1 | **0.0** | 0.3 | **0.0** |
| ER5 | V | 368.7 | 139.6 | 73.8 | 33.2 | 21.5 | 14.0 | 9.5 | 7.0 | 5.8 | 1.5 | **0.6** | **0.0** | **0.0** |
| | S | 349.0 | 139.0 | 81.2 | 45.7 | 40.7 | 27.6 | 22.1 | 22.7 | 24.7 | 17.6 | 14.0 | 19.1 | 18.0 |
| | VS | 426.5 | 177.1 | 87.8 | 36.9 | 21.5 | 15.7 | 11.2 | 12.3 | 5.7 | 2.6 | 1.9 | 0.1 | **0.0** |
| | V+ | **83.0** | **44.2** | **27.8** | **17.5** | 12.5 | **9.2** | **7.0** | **5.1** | **4.7** | **1.3** | 0.7 | **0.0** | 0.1 |
| | S+ | 95.4 | 56.3 | 40.4 | 30.2 | 32.9 | 23.4 | 19.7 | 18.1 | 22.6 | 17.3 | 13.5 | 19.3 | 18.0 |
| | VS+ | 88.6 | 47.0 | 27.9 | 18.0 | **11.5** | **9.2** | 8.0 | 7.1 | 5.4 | 1.7 | 2.2 | **0.0** | 0.1 |
| SF5 | V | 92.6 | 31.9 | **12.6** | **6.0** | **4.9** | **3.9** | **2.5** | **1.8** | **1.6** | **0.3** | 0.4 | **0.0** | **0.0** |
| | S | 81.2 | 32.1 | 21.8 | 13.4 | 16.9 | 13.5 | 12.6 | 12.3 | 11.4 | 10.1 | 8.9 | 12.3 | 11.6 |
| | VS | 167.9 | 63.9 | 39.1 | 8.5 | 11.5 | 4.0 | 3.2 | 2.0 | 2.1 | **0.3** | 5.0 | **0.0** | 1.9 |
| | V+ | **40.3** | **21.0** | 15.6 | 8.3 | 7.2 | 5.3 | 4.1 | 2.3 | 2.8 | 0.5 | **0.4** | **0.0** | **0.0** |
| | S+ | 49.3 | 29.0 | 22.8 | 16.0 | 19.3 | 15.3 | 13.9 | 13.1 | 12.8 | 11.8 | 8.9 | 12.4 | 11.6 |
| | VS+ | 47.6 | 23.2 | 16.4 | 9.1 | 7.4 | 5.6 | 4.5 | 2.5 | 3.3 | 0.5 | 4.9 | **0.0** | 1.9 |

Table B.5: SHD results on 100 nodes linear Gumbel noise SEM with different sample size.

| | | 200 | 300 | 400 | 500 | 600 | 700 | 800 | 900 | 1000 | 1500 | 2000 | 2500 | 3000 |
|---|---|---|---|---|---|---|---|---|---|---|---|---|---|---|
| ER2 | V | 124.7 | 35.1 | 11.8 | 6.5 | 3.6 | 2.2 | **0.8** | 0.6 | **0.9** | 0.4 | **0.0** | 0.1 | **0.0** |
| | S | 94.2 | 39.8 | 22.3 | 18.2 | 15.7 | 15.4 | 17.4 | 15.7 | 19.7 | 18.0 | 18.6 | 20.9 | 22.3 |
| | VS | 145.1 | 55.6 | 20.2 | 10.6 | 22.9 | 5.4 | 12.4 | 3.2 | 1.9 | **0.2** | 0.3 | 7.2 | 1.1 |
| | V+ | 39.8 | **15.6** | **7.3** | **4.0** | **3.0** | **1.5** | 1.2 | **0.5** | **0.9** | 0.4 | **0.0** | 0.1 | **0.0** |
| | S+ | **38.5** | 28.1 | 19.7 | 16.2 | 16.1 | 14.8 | 19.1 | 15.7 | 20.3 | 17.7 | 19.7 | 21.0 | 22.3 |
| | VS+ | 68.4 | 30.0 | 13.6 | 6.5 | 17.1 | 4.3 | 12.3 | 0.8 | 1.1 | 0.4 | 0.5 | 7.1 | 1.1 |
| SF2 | V | 46.9 | 8.6 | 1.9 | **0.8** | **0.6** | **0.3** | 0.2 | **0.3** | 0.1 | **0.0** | **0.0** | **0.0** | **0.0** |
| | S | 27.7 | 11.3 | 3.5 | 5.4 | 3.7 | 3.3 | 2.9 | 4.0 | 4.8 | 5.6 | 5.6 | 5.3 | 5.5 |
| | VS | 58.4 | 24.6 | 3.4 | 3.4 | 1.1 | 1.4 | **0.2** | 1.3 | 1.1 | 0.1 | 0.1 | 0.4 | **0.0** |
| | V+ | **12.0** | **4.5** | **1.5** | **0.8** | **0.6** | **0.3** | 0.2 | 0.4 | **0.1** | **0.0** | **0.0** | **0.0** | **0.0** |
| | S+ | 12.8 | 8.4 | 3.2 | 5.3 | 3.6 | 3.4 | 2.8 | 4.2 | 4.7 | 5.6 | 5.6 | 5.3 | 5.5 |
| | VS+ | 14.7 | 7.2 | 3.9 | 1.4 | 1.2 | 1.5 | **0.2** | 0.8 | 3.1 | 0.1 | 0.1 | 0.4 | **0.0** |
| ER5 | V | 362.8 | 139.7 | 70.9 | 40.2 | 23.2 | 17.4 | 9.6 | 5.9 | 6.1 | 1.6 | **0.4** | **0.3** | 0.1 |
| | S | 338.2 | 138.0 | 84.0 | 53.2 | 37.7 | 31.5 | 21.3 | 28.0 | 21.2 | 16.8 | 16.1 | 19.5 | 16.6 |
| | VS | 448.6 | 199.8 | 93.9 | 58.3 | 26.3 | 21.8 | 13.7 | 44.1 | 22.3 | **1.3** | 0.8 | 0.8 | 3.8 |
| | V+ | **119.1** | **64.7** | **35.9** | **21.6** | **13.3** | **10.7** | **6.8** | **5.5** | **4.7** | 1.5 | 0.5 | **0.3** | 0.1 |
| | S+ | 128.2 | 72.7 | 49.3 | 36.4 | 28.5 | 27.3 | 19.2 | 24.9 | 18.8 | 17.2 | 16.1 | 17.5 | 16.4 |
| | VS+ | 133.6 | 82.8 | 37.3 | 29.0 | 14.6 | 14.0 | 6.9 | 41.4 | 20.6 | 1.4 | 0.7 | 0.7 | 3.6 |
| SF5 | V | 104.4 | 31.9 | 12.0 | **7.6** | **5.2** | **2.9** | **2.4** | **2.1** | 2.7 | **0.5** | **0.2** | 0.1 | **0.0** |
| | S | 82.8 | 32.2 | 20.3 | 14.6 | 14.5 | 10.7 | 11.7 | 12.3 | 9.4 | 10.3 | 13.5 | 11.2 | 11.1 |
| | VS | 177.6 | 95.0 | 31.1 | 21.4 | 12.4 | 5.5 | 4.5 | 7.9 | 20.3 | 0.6 | 0.5 | **0.1** | **0.0** |
| | V+ | 47.2 | **20.2** | **11.0** | 8.1 | 5.8 | 3.6 | 3.3 | 2.2 | **1.9** | **0.5** | 0.3 | **0.1** | **0.0** |
| | S+ | **45.9** | 26.3 | 20.0 | 15.6 | 16.0 | 11.7 | 12.9 | 12.6 | 10.4 | 11.8 | 13.6 | 11.3 | 11.1 |
| | VS+ | 48.3 | 23.4 | 17.9 | 8.8 | 6.8 | 5.6 | 4.6 | 4.9 | 8.5 | 0.8 | 0.6 | **0.1** | **0.0** |

Table B.6: SHD results on 100 nodes linear Exponential noise SEM with different sample size.

| | | 200 | 300 | 400 | 500 | 600 | 700 | 800 | 900 | 1000 | 1500 | 2000 | 2500 | 3000 |
|---|---|---|---|---|---|---|---|---|---|---|---|---|---|---|
| ER2 | V | 132.1 | 39.0 | 15.3 | 5.9 | 3.7 | **2.5** | 1.3 | **0.6** | 0.7 | **0.2** | **0.0** | 0.2 | 0.0 |
| | S | 106.8 | 34.4 | 20.3 | 16.4 | 15.0 | 17.7 | 13.9 | 20.5 | 19.0 | 19.9 | 15.3 | 23.5 | 22.2 |
| | VS | 138.0 | 53.8 | 23.6 | 12.4 | 6.1 | 3.5 | 4.5 | 5.5 | 14.1 | **0.2** | 3.7 | 0.4 | 0.2 |
| | V+ | **31.8** | **10.7** | **7.5** | **3.4** | **2.0** | 2.5 | **1.2** | 0.8 | **0.7** | **0.2** | **0.0** | 0.2 | 0.0 |
| | S+ | 34.0 | 20.2 | 17.2 | 15.9 | 13.7 | 16.2 | 13.8 | 19.7 | 19.2 | 20.0 | 15.3 | 22.8 | 22.1 |
| | VS+ | 44.1 | 22.1 | 12.8 | 6.7 | 3.7 | 2.8 | 3.8 | 5.6 | 13.2 | 0.3 | 4.0 | 0.6 | 0.4 |
| SF2 | V | 55.4 | 17.0 | 5.1 | 2.5 | **0.5** | 1.6 | 0.9 | **0.1** | 0.3 | **0.0** | **0.0** | **0.0** | **0.0** |
| | S | 33.9 | 11.4 | 4.3 | 6.0 | 4.7 | 6.1 | 4.4 | 4.2 | 4.3 | 4.1 | 4.1 | 4.9 | 5.7 |
| | VS | 67.5 | 16.2 | 8.9 | 5.0 | 1.5 | 1.0 | **0.6** | 0.7 | 0.3 | 0.1 | 0.7 | **0.0** | **0.0** |
| | V+ | **12.1** | **3.5** | **3.0** | **0.8** | 0.8 | **0.5** | 0.9 | **0.1** | 0.2 | 0.1 | **0.0** | **0.0** | **0.0** |
| | S+ | 14.4 | 10.5 | 4.3 | 5.8 | 5.0 | 6.1 | 4.4 | 5.0 | 4.3 | 4.1 | 4.1 | 4.9 | 5.6 |
| | VS+ | 17.2 | 5.6 | 3.8 | 2.7 | 2.2 | 1.0 | **0.6** | **0.1** | 0.2 | 0.2 | 0.7 | **0.0** | **0.0** |
| ER5 | V | 393.8 | 157.8 | 75.7 | 40.5 | 29.6 | 18.4 | 10.2 | 8.9 | 5.2 | 1.6 | **0.9** | 0.2 | 0.0 |
| | S | 361.2 | 154.7 | 80.5 | 51.2 | 44.3 | 34.6 | 28.3 | 19.3 | 24.3 | 16.2 | 16.8 | 18.1 | 15.7 |
| | VS | 474.7 | 193.9 | 110.3 | 62.5 | 35.6 | 32.1 | 20.0 | 26.4 | 51.6 | 2.5 | 1.3 | 12.0 | 3.1 |
| | V+ | 96.6 | 56.6 | **32.2** | **19.6** | **15.3** | **12.8** | **8.1** | **7.1** | **5.1** | **1.4** | 1.1 | **0.1** | 0.0 |
| | S+ | **95.8** | 66.9 | 41.4 | 33.2 | 32.0 | 27.2 | 23.2 | 20.1 | 21.0 | 14.4 | 16.6 | 19.7 | 14.4 |
| | VS+ | 103.8 | **55.8** | 40.0 | 26.1 | 15.8 | 15.5 | 10.6 | 19.3 | 48.1 | 2.0 | 1.0 | 7.3 | 3.1 |
| SF5 | V | 108.9 | 34.9 | 23.1 | **7.5** | 11.6 | **3.2** | **2.7** | **1.6** | 0.6 | 0.6 | **0.1** | **0.0** | 0.0 |
| | S | 84.8 | 34.8 | 21.7 | 13.5 | 16.6 | 11.4 | 13.0 | 12.2 | 8.8 | 11.6 | 12.2 | 8.8 | 11.3 |
| | VS | 203.5 | 80.3 | 35.2 | 11.0 | **5.4** | 10.4 | 9.6 | 3.4 | **0.6** | 0.7 | 4.6 | 2.3 | **0.0** |
| | V+ | **51.0** | 27.9 | **15.5** | 9.1 | 7.1 | 4.7 | 4.8 | 2.6 | 2.4 | 1.2 | 0.2 | **0.0** | 0.0 |
| | S+ | 54.4 | 34.0 | 24.8 | 16.3 | 19.5 | 12.5 | 15.4 | 13.1 | 11.0 | 12.1 | 12.3 | 8.8 | 11.3 |
| | VS+ | 56.9 | **26.3** | 17.2 | 10.5 | 7.2 | 5.5 | 8.1 | 3.5 | 2.4 | 1.1 | 4.7 | 2.0 | **0.0** |

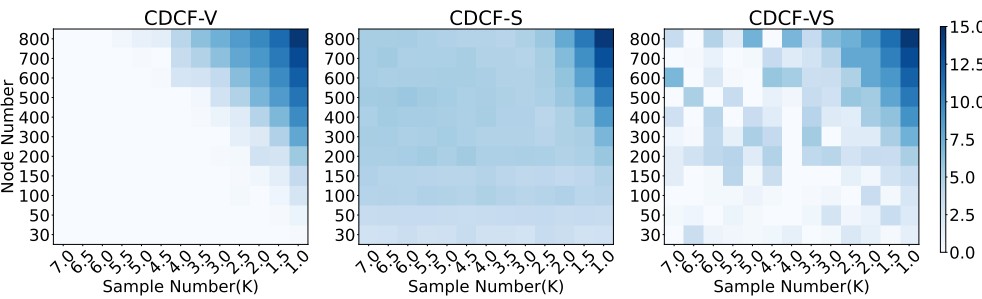

Figure B.4: Performance (log(SHD + 1)) upper bound of CDCF-* on ER2 Gaussian.

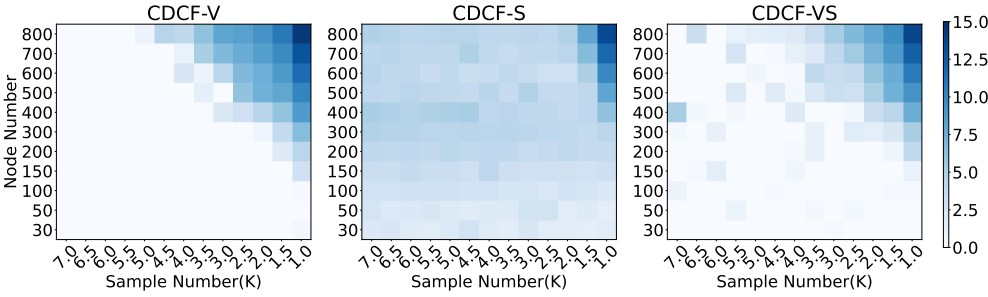

Figure B.5: Performance (log(SHD + 1)) upper bound of CDCF-* on SF2 Gaussian.

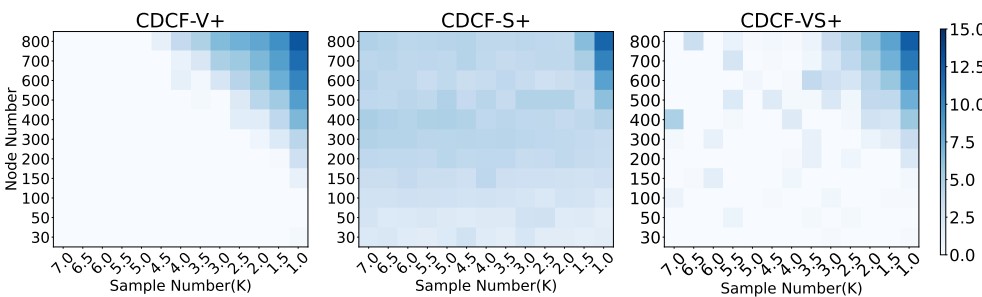

Figure B.6: Performance (log(SHD + 1)) upper bound of CDCF-*+ on SF2 Gaussian.

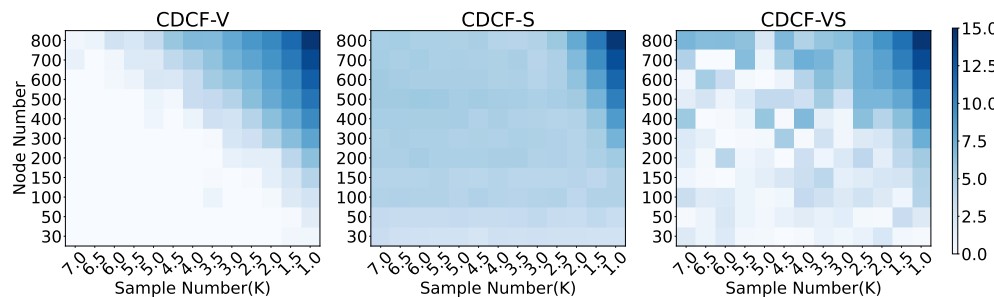

Figure B.7: Performance (log(SHD + 1)) upper bound of CDCF-* on ER2 Exponential.

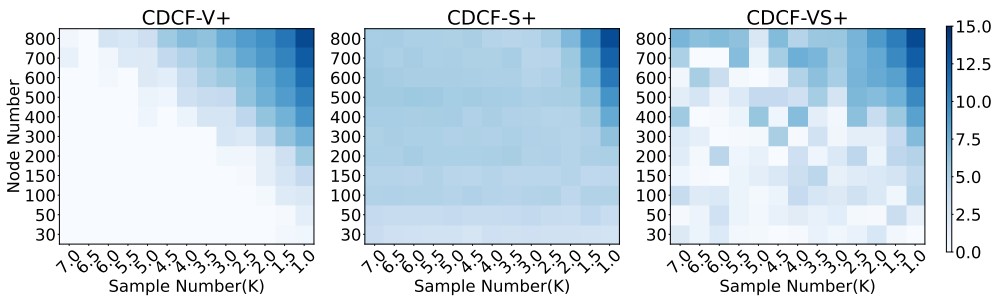

Figure B.8: Performance (log(SHD + 1)) upper bound of CDCF-*+ on ER2 Exponential.

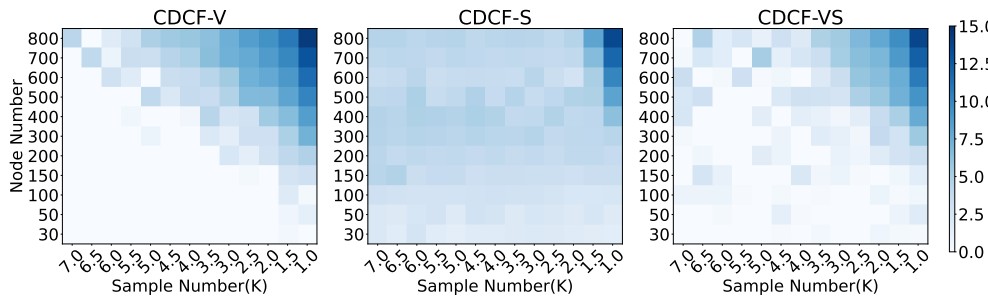

Figure B.9: Performance (log(SHD + 1)) upper bound of CDCF-* on SF2 Exponential.

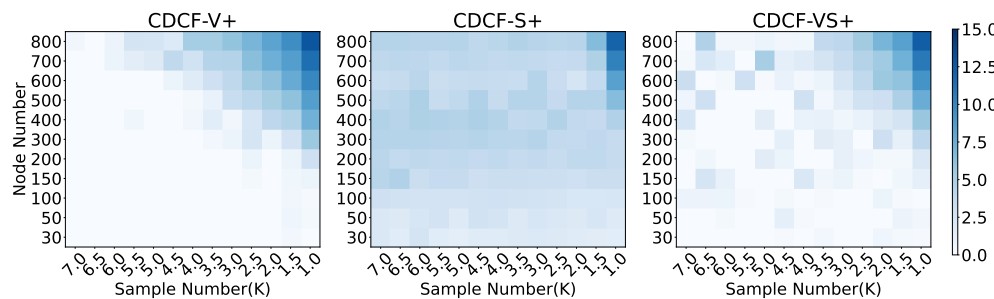

Figure B.10: Performance (log(SHD + 1)) upper bound of CDCF-*+ on SF2 Exponential.

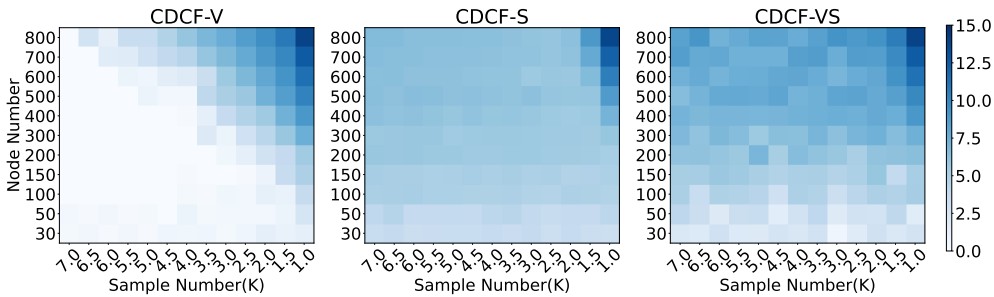

Figure B.11: Performance (log(SHD + 1)) upper bound of CDCF-* on ER2 Gumbel.

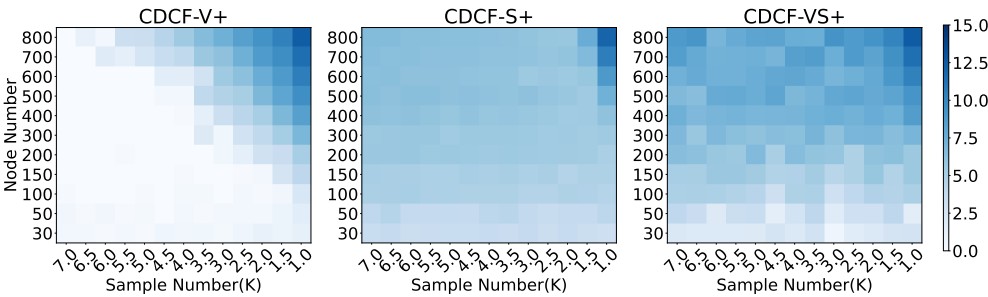

Figure B.12: Performance (log(SHD + 1)) upper bound of CDCF-*+ on ER2 Gumbel.

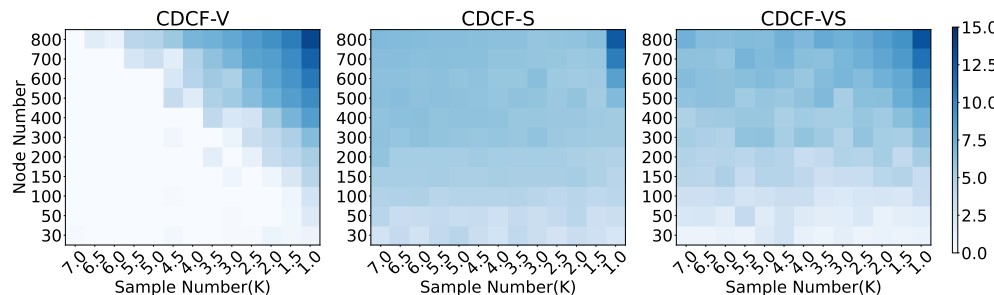

Figure B.13: Performance (log(SHD + 1)) upper bound of CDCF-* on SF2 Gumbel.

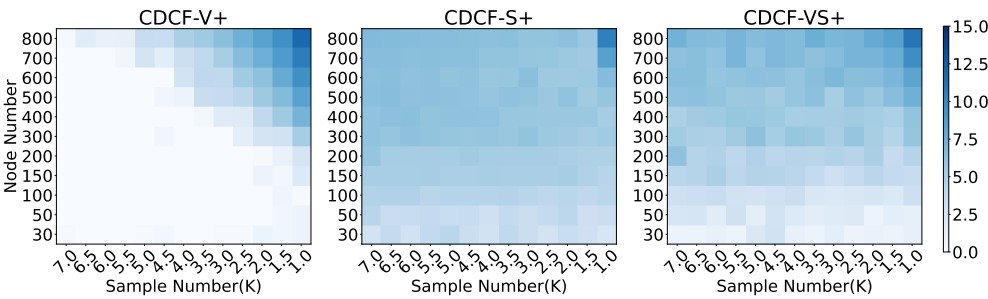

Figure B.14: Performance (log(SHD + 1)) upper bound of CDCF-*+ on SF2 Gumbel.

Table B.7: CDCF-V+ SHD results on different $\gamma$ with different sample size on 100 nodes linear Gaussian SEM.

| CDCF-V+ | 200 | 300 | 400 | 500 | 600 | 700 | 800 | 900 | 1000 | 1500 | 2000 | 2500 | 3000 |
|---|---|---|---|---|---|---|---|---|---|---|---|---|---|
| **ER2** | | | | | | | | | | | | | |
| $\gamma = 0.0$ | 114.3 | 32.2 | 11.7 | 4.6 | 2.5 | 1.4 | **1.1** | 0.5 | 0.4 | 0.3 | 0.0 | 0.0 | 0.0 |
| $\gamma = 1.0$ | 21.9 | 7.9 | 4.3 | **2.8** | **2.0** | **1.3** | 1.4 | 0.5 | 0.4 | 0.3 | 0.0 | 0.0 | 0.0 |
| $\gamma = 2.0$ | **13.6** | **7.2** | **4.0** | 2.9 | 2.1 | 1.7 | 1.4 | 0.9 | 0.7 | 0.3 | 0.0 | 0.0 | 0.0 |
| $\gamma = 3.0$ | 14.8 | 8.1 | 4.5 | 3.5 | 2.7 | 2.0 | 1.8 | 1.4 | 0.7 | 0.3 | 0.0 | 0.0 | 0.0 |
| $\gamma = 4.0$ | 18.3 | 9.9 | 7.2 | 4.9 | 3.1 | 3.0 | 2.3 | 1.4 | 0.8 | 0.6 | 0.0 | 0.0 | 0.0 |
| $\gamma = 5.0$ | 23.0 | 12.3 | 9.3 | 6.5 | 4.1 | 3.8 | 2.4 | 1.6 | 1.0 | 0.8 | 0.1 | 0.0 | 0.0 |
| $\gamma = 6.0$ | 27.0 | 14.9 | 11.4 | 7.7 | 4.5 | 4.4 | 3.6 | 1.9 | 1.7 | 0.8 | 0.1 | 0.1 | 0.0 |
| $\gamma = 7.0$ | 33.0 | 19.1 | 13.7 | 9.1 | 6.0 | 4.9 | 4.3 | 2.5 | 2.4 | 0.9 | 0.2 | 0.1 | 0.0 |
| $\gamma = 8.0$ | 37.3 | 22.1 | 15.5 | 10.2 | 7.0 | 5.7 | 5.1 | 2.9 | 3.1 | 1.2 | 0.2 | 0.1 | 0.0 |
| $\gamma = 9.0$ | 42.5 | 25.1 | 17.6 | 11.6 | 8.6 | 6.9 | 5.8 | 3.6 | 3.4 | 1.2 | 0.2 | 0.1 | 0.1 |
| $\gamma = 10.0$ | 46.7 | 27.4 | 20.0 | 12.8 | 10.0 | 7.5 | 6.6 | 4.5 | 3.7 | 1.3 | 0.3 | 0.1 | 0.1 |
| **SF2** | | | | | | | | | | | | | |
| $\gamma = 0.0$ | 37.1 | 7.2 | 1.9 | 2.6 | **0.9** | **0.3** | **0.1** | **0.1** | 0.2 | 0.0 | 0.0 | 0.0 | 0.0 |
| $\gamma = 1.0$ | **8.8** | **3.0** | **1.6** | **0.7** | 1.1 | 0.5 | 0.2 | 0.2 | **0.1** | 0.0 | 0.0 | 0.0 | 0.0 |
| $\gamma = 2.0$ | 9.7 | 4.2 | 2.2 | 0.9 | 1.2 | 0.5 | 0.2 | **0.1** | **0.1** | 0.0 | 0.0 | 0.0 | 0.0 |
| $\gamma = 3.0$ | 11.5 | 6.2 | 2.9 | 1.4 | 1.2 | 0.4 | 0.3 | **0.1** | **0.1** | 0.0 | 0.0 | 0.0 | 0.0 |
| $\gamma = 4.0$ | 15.4 | 7.5 | 3.5 | 1.8 | 1.6 | 0.8 | 0.4 | 0.3 | 0.2 | 0.0 | 0.0 | 0.0 | 0.0 |
| $\gamma = 5.0$ | 18.1 | 8.9 | 4.3 | 2.7 | 2.1 | 1.3 | 0.5 | 0.5 | 0.4 | 0.0 | 0.0 | 0.0 | 0.0 |
| $\gamma = 6.0$ | 20.9 | 10.8 | 5.4 | 3.3 | 2.5 | 1.7 | 0.8 | 1.0 | 0.6 | 0.0 | 0.0 | 0.0 | 0.0 |
| $\gamma = 7.0$ | 23.5 | 13.1 | 6.7 | 5.2 | 4.9 | 2.7 | 1.5 | 1.3 | 0.7 | 0.0 | 0.0 | 0.0 | 0.0 |
| $\gamma = 8.0$ | 27.0 | 14.8 | 8.1 | 5.8 | 6.0 | 2.9 | 2.2 | 1.5 | 0.7 | 0.0 | 0.0 | 0.0 | 0.0 |
| $\gamma = 9.0$ | 29.4 | 16.8 | 9.8 | 6.7 | 6.7 | 3.5 | 2.7 | 2.0 | 1.1 | 0.0 | 0.0 | 0.0 | 0.0 |
| $\gamma = 10.0$ | 32.0 | 19.0 | 11.5 | 7.4 | 7.7 | 4.1 | 3.0 | 2.3 | 1.5 | 0.1 | 0.0 | 0.0 | 0.0 |
| **ER5** | | | | | | | | | | | | | |
| $\gamma = 0.0$ | 368.7 | 139.6 | 73.8 | 33.2 | 21.5 | 14.0 | 9.5 | 7.0 | 5.8 | 1.5 | **0.6** | 0.0 | 0.0 |
| $\gamma = 1.0$ | 83.0 | 44.2 | 27.8 | **17.5** | **12.5** | **9.2** | **7.0** | **5.1** | **4.7** | **1.3** | 0.7 | 0.0 | 0.1 |
| $\gamma = 2.0$ | **74.1** | **41.7** | **26.2** | 19.8 | 13.4 | 10.7 | 8.4 | 6.6 | 5.7 | **1.3** | 1.1 | 0.0 | 0.3 |
| $\gamma = 3.0$ | 82.1 | 52.3 | 33.1 | 23.0 | 18.2 | 14.5 | 11.9 | 8.0 | 7.2 | 2.4 | 1.6 | 0.2 | 0.4 |
| $\gamma = 4.0$ | 94.8 | 62.1 | 40.1 | 28.6 | 23.9 | 18.1 | 14.3 | 10.3 | 10.3 | 3.9 | 2.1 | 0.4 | 0.3 |
| $\gamma = 5.0$ | 109.7 | 72.3 | 51.4 | 37.6 | 30.1 | 22.6 | 17.6 | 14.1 | 12.8 | 5.6 | 2.6 | 0.8 | 0.4 |
| $\gamma = 6.0$ | 123.3 | 84.2 | 57.9 | 47.8 | 34.7 | 26.3 | 21.2 | 17.4 | 15.4 | 6.7 | 3.2 | 0.9 | 0.5 |
| $\gamma = 7.0$ | 140.3 | 100.2 | 67.9 | 53.1 | 39.9 | 31.2 | 25.0 | 20.2 | 18.2 | 7.8 | 3.7 | 1.0 | 0.7 |
| $\gamma = 8.0$ | 152.7 | 113.0 | 74.8 | 61.5 | 47.5 | 36.4 | 31.0 | 23.5 | 22.3 | 9.4 | 4.4 | 1.4 | 0.9 |
| $\gamma = 9.0$ | 162.5 | 122.0 | 83.1 | 67.7 | 51.6 | 42.3 | 34.6 | 28.5 | 25.7 | 10.7 | 4.9 | 1.9 | 1.4 |
| $\gamma = 10.0$ | 175.5 | 130.2 | 92.6 | 73.8 | 59.0 | 48.8 | 37.7 | 33.2 | 28.7 | 12.5 | 5.7 | 2.4 | 1.8 |
| **SF5** | | | | | | | | | | | | | |
| $\gamma = 0.0$ | 92.6 | 31.9 | **12.6** | **6.0** | **4.9** | **3.9** | **2.5** | 1.8 | **1.6** | 0.3 | 0.4 | 0.0 | 0.0 |
| $\gamma = 1.0$ | **40.3** | **21.0** | 15.6 | 8.3 | 7.2 | 5.3 | 4.1 | 2.3 | 2.8 | 0.5 | 0.4 | 0.0 | 0.0 |
| $\gamma = 2.0$ | 57.2 | 35.7 | 23.3 | 16.1 | 11.2 | 9.6 | 7.5 | 5.9 | 5.1 | 1.7 | 0.6 | 0.0 | 0.0 |
| $\gamma = 3.0$ | 69.8 | 46.2 | 31.3 | 23.6 | 16.9 | 12.5 | 11.2 | 8.6 | 7.0 | 2.8 | 1.4 | 0.0 | 0.1 |
| $\gamma = 4.0$ | 80.9 | 53.1 | 38.7 | 30.4 | 21.1 | 17.9 | 14.5 | 11.9 | 8.4 | 4.2 | 2.1 | 0.9 | 0.2 |
| $\gamma = 5.0$ | 90.6 | 66.3 | 43.7 | 36.3 | 25.8 | 24.5 | 20.3 | 16.6 | 12.1 | 5.7 | 3.3 | 1.6 | 0.4 |
| $\gamma = 6.0$ | 103.3 | 73.3 | 49.1 | 42.3 | 34.6 | 28.1 | 25.5 | 19.7 | 14.4 | 8.3 | 4.0 | 2.4 | 1.5 |
| $\gamma = 7.0$ | 110.3 | 79.4 | 55.9 | 45.6 | 38.0 | 31.6 | 29.1 | 24.6 | 19.9 | 10.3 | 5.1 | 3.6 | 2.1 |
| $\gamma = 8.0$ | 118.4 | 86.1 | 63.6 | 50.2 | 41.3 | 35.7 | 33.5 | 27.6 | 21.4 | 12.6 | 5.8 | 4.3 | 2.5 |
| $\gamma = 9.0$ | 124.0 | 92.8 | 68.7 | 54.1 | 46.5 | 39.4 | 36.3 | 30.2 | 25.0 | 13.9 | 7.2 | 4.7 | 3.2 |
| $\gamma = 10.0$ | 129.5 | 98.6 | 74.3 | 57.8 | 49.6 | 42.2 | 39.3 | 32.6 | 27.1 | 17.5 | 7.7 | 5.9 | 4.0 |

Table B.8: CDCF-S+ SHD results on different $\gamma$ with different sample size on 100 nodes linear Gaussian SEM.

| | CDCF-S+ | 200 | 300 | 400 | 500 | 600 | 700 | 800 | 900 | 1000 | 1500 | 2000 | 2500 | 3000 |
|---|---|---|---|---|---|---|---|---|---|---|---|---|---|---|
| ER2 | $\gamma = 0.0$ | 98.4 | 30.3 | 18.6 | 18.4 | 15.3 | 18.5 | 18.2 | 16.0 | 15.4 | **16.5** | **18.1** | 21.9 | 18.5 |
| | $\gamma = 1.0$ | 29.1 | 17.6 | 13.5 | **17.2** | **14.6** | **16.3** | 18.5 | 16.3 | **15.3** | **16.5** | **18.1** | 21.8 | 18.5 |
| | $\gamma = 2.0$ | **22.8** | **16.8** | **13.0** | 17.4 | 15.3 | 17.0 | **18.0** | 16.4 | 15.4 | 16.6 | 18.2 | 22.3 | 18.5 |
| | $\gamma = 3.0$ | 26.9 | 18.4 | 15.6 | 17.9 | 15.6 | 18.0 | 18.4 | 16.1 | 16.8 | 17.4 | **18.1** | **21.6** | 18.4 |
| | $\gamma = 4.0$ | 32.5 | 21.6 | 17.5 | 18.1 | 15.9 | 19.2 | 19.0 | **15.9** | 17.1 | 17.7 | 18.3 | 22.1 | 18.4 |
| | $\gamma = 5.0$ | 36.1 | 23.9 | 18.8 | 19.1 | 16.5 | 20.6 | 19.5 | **15.9** | 17.0 | 17.6 | **18.1** | 22.3 | 18.6 |
| | $\gamma = 6.0$ | 39.8 | 26.4 | 20.6 | 21.6 | 17.9 | 21.6 | 20.2 | 16.8 | 17.4 | 17.8 | 18.6 | 22.2 | 18.6 |
| | $\gamma = 7.0$ | 43.9 | 28.8 | 22.9 | 22.4 | 19.4 | 22.0 | 21.1 | 17.4 | 19.3 | 18.4 | 18.6 | 22.2 | 18.4 |
| | $\gamma = 8.0$ | 48.3 | 32.5 | 24.9 | 25.0 | 19.7 | 22.1 | 22.0 | 18.6 | 19.5 | 18.4 | 19.1 | 21.8 | **18.3** |
| | $\gamma = 9.0$ | 52.1 | 34.3 | 27.5 | 25.5 | 20.8 | 22.7 | 23.2 | 19.1 | 20.5 | 18.7 | 18.7 | 21.8 | **18.3** |
| | $\gamma = 10.0$ | 56.1 | 37.7 | 29.7 | 26.4 | 22.2 | 23.3 | 24.1 | 19.7 | 20.5 | 18.7 | 18.8 | 22.5 | 19.0 |
| SF2 | $\gamma = 0.0$ | 27.3 | 7.5 | **6.9** | 3.8 | **4.2** | 4.9 | **3.9** | **3.3** | **3.6** | 5.3 | 5.7 | **3.4** | **4.8** |
| | $\gamma = 1.0$ | **11.7** | **4.8** | 7.3 | **3.6** | 4.8 | 5.0 | 4.0 | **3.3** | 4.4 | 5.3 | 5.7 | **3.4** | **4.8** |
| | $\gamma = 2.0$ | 13.6 | 5.8 | 8.0 | 4.6 | 5.0 | 5.2 | **3.9** | 3.8 | 4.3 | **5.2** | 5.7 | **3.4** | **4.8** |
| | $\gamma = 3.0$ | 14.8 | 6.9 | 7.9 | 4.5 | 5.2 | 5.3 | 4.0 | 3.8 | 4.3 | **5.2** | 5.7 | **3.4** | 4.9 |
| | $\gamma = 4.0$ | 17.9 | 8.5 | 8.3 | 5.4 | 6.1 | 5.5 | 4.2 | 4.0 | 4.3 | **5.2** | 5.7 | **3.4** | 4.9 |
| | $\gamma = 5.0$ | 20.4 | 9.8 | 9.0 | 6.4 | 6.6 | 5.9 | 4.3 | 3.9 | 5.1 | 5.3 | **5.6** | **3.4** | 4.9 |
| | $\gamma = 6.0$ | 24.0 | 11.3 | 10.3 | 7.0 | 6.9 | 6.4 | 4.4 | 4.4 | 5.9 | 5.3 | **5.6** | **3.4** | 4.9 |
| | $\gamma = 7.0$ | 26.8 | 13.4 | 11.6 | 7.4 | 7.6 | 6.6 | 5.1 | 5.7 | 6.2 | 5.3 | **5.6** | **3.4** | **4.8** |
| | $\gamma = 8.0$ | 29.9 | 15.0 | 12.3 | 7.9 | 8.5 | 6.8 | 5.8 | 5.9 | 6.2 | 5.3 | **5.6** | **3.4** | **4.8** |
| | $\gamma = 9.0$ | 31.9 | 17.2 | 14.0 | 9.2 | 10.3 | 7.5 | 6.3 | 6.4 | 6.5 | 5.3 | **5.6** | **3.4** | **4.8** |
| | $\gamma = 10.0$ | 34.2 | 19.4 | 15.4 | 9.9 | 11.5 | 8.0 | 7.0 | 6.5 | 7.1 | 5.4 | **5.6** | **3.4** | **4.8** |
| ER5 | $\gamma = 0.0$ | 349.0 | 139.0 | 81.2 | 45.7 | 40.7 | 27.6 | 22.1 | 22.7 | 24.7 | 17.6 | 14.0 | **19.1** | **18.0** |
| | $\gamma = 1.0$ | 95.4 | 56.3 | **40.4** | **30.2** | **32.9** | **23.4** | **19.7** | **18.1** | **22.6** | **17.3** | **13.5** | 19.3 | **18.0** |
| | $\gamma = 2.0$ | **86.1** | **55.9** | 40.8 | 38.6 | 34.0 | 23.5 | 21.6 | 20.2 | 24.7 | **17.3** | 14.8 | **19.1** | 18.2 |
| | $\gamma = 3.0$ | 94.2 | 66.0 | 49.6 | 42.9 | 40.4 | 29.4 | 25.5 | 25.6 | 25.5 | 18.1 | 16.1 | 19.3 | 18.2 |
| | $\gamma = 4.0$ | 105.8 | 77.2 | 57.8 | 48.0 | 47.5 | 33.7 | 29.2 | 27.1 | 28.0 | 21.8 | 16.9 | 20.2 | 18.1 |
| | $\gamma = 5.0$ | 119.9 | 89.8 | 64.9 | 56.1 | 48.1 | 38.5 | 33.1 | 31.9 | 31.2 | 23.3 | 17.9 | 20.5 | 18.2 |
| | $\gamma = 6.0$ | 132.9 | 100.3 | 74.0 | 64.9 | 53.8 | 44.5 | 39.1 | 35.3 | 36.0 | 24.7 | 19.1 | 20.8 | 19.2 |
| | $\gamma = 7.0$ | 147.3 | 110.8 | 80.5 | 71.3 | 57.5 | 49.7 | 42.7 | 40.4 | 39.3 | 26.5 | 22.1 | 25.7 | 20.2 |
| | $\gamma = 8.0$ | 159.9 | 121.1 | 88.9 | 77.1 | 64.8 | 54.9 | 46.7 | 44.0 | 42.1 | 29.0 | 23.0 | 26.0 | 20.5 |
| | $\gamma = 9.0$ | 171.5 | 129.4 | 98.2 | 83.3 | 69.3 | 60.9 | 50.0 | 48.3 | 45.0 | 30.6 | 23.4 | 26.8 | 21.6 |
| | $\gamma = 10.0$ | 185.9 | 138.5 | 105.0 | 87.6 | 73.9 | 64.7 | 54.8 | 52.2 | 48.5 | 31.9 | 23.9 | 27.1 | 23.5 |
| SF5 | $\gamma = 0.0$ | 81.2 | 32.1 | **21.8** | **13.4** | **16.9** | **13.5** | **12.6** | **12.3** | 11.4 | 10.1 | **8.9** | 12.3 | **11.6** |
| | $\gamma = 1.0$ | **49.3** | **29.0** | 22.8 | 16.0 | 19.3 | 15.3 | 13.9 | 13.1 | 12.8 | 11.8 | **8.9** | 12.4 | **11.6** |
| | $\gamma = 2.0$ | 59.6 | 38.5 | 29.3 | 22.3 | 23.1 | 19.5 | 17.5 | 16.9 | 15.4 | 13.2 | 9.3 | 12.1 | **11.6** |
| | $\gamma = 3.0$ | 73.4 | 48.9 | 35.9 | 28.2 | 27.6 | 22.3 | 20.6 | 19.6 | 17.5 | 14.3 | 10.5 | 12.1 | 11.7 |
| | $\gamma = 4.0$ | 82.7 | 55.6 | 42.9 | 34.1 | 31.4 | 26.3 | 24.2 | 23.6 | 19.0 | 15.6 | 11.2 | **11.7** | 11.9 |
| | $\gamma = 5.0$ | 93.2 | 65.0 | 48.2 | 39.2 | 36.2 | 30.3 | 27.4 | 26.6 | 21.2 | 17.0 | 12.4 | 12.4 | 12.1 |
| | $\gamma = 6.0$ | 102.7 | 71.7 | 53.3 | 44.9 | 41.0 | 34.0 | 30.4 | 28.4 | 23.5 | 18.3 | 14.1 | 13.2 | 13.3 |
| | $\gamma = 7.0$ | 109.3 | 78.0 | 60.7 | 48.4 | 44.4 | 37.8 | 33.9 | 32.0 | 25.6 | 20.3 | 16.6 | 14.4 | 13.9 |
| | $\gamma = 8.0$ | 117.5 | 84.3 | 67.3 | 53.3 | 48.6 | 40.7 | 36.7 | 34.9 | 27.0 | 21.1 | 17.4 | 15.1 | 14.3 |
| | $\gamma = 9.0$ | 123.2 | 90.9 | 72.2 | 56.9 | 51.6 | 44.2 | 40.4 | 37.5 | 31.0 | 22.4 | 18.8 | 15.6 | 15.0 |
| | $\gamma = 10.0$ | 129.8 | 95.9 | 77.6 | 61.0 | 54.4 | 46.9 | 43.4 | 39.8 | 33.2 | 24.0 | 19.4 | 16.8 | 15.8 |

Table B.9: CDCF-VS+ SHD results on different $\gamma$ with different sample size on 100 nodes linear Gaussian SEM.

| | CDCF-VS+ | 200 | 300 | 400 | 500 | 600 | 700 | 800 | 900 | 1000 | 1500 | 2000 | 2500 | 3000 |
|---|---|---|---|---|---|---|---|---|---|---|---|---|---|---|
| ER2 | $\gamma = 0.0$ | 141.6 | 38.7 | 32.3 | 6.1 | 8.0 | 12.5 | 2.5 | 8.6 | 0.6 | **0.3** | 1.2 | **0.0** | 1.0 |
| | $\gamma = 1.0$ | 38.6 | 16.0 | 21.3 | **3.0** | 4.3 | 10.4 | 2.1 | 8.1 | **0.4** | **0.3** | 0.8 | **0.0** | 0.9 |
| | $\gamma = 2.0$ | **18.4** | **9.6** | 16.5 | 3.3 | **3.6** | 13.0 | 2.4 | 8.1 | 0.7 | 0.5 | **0.0** | **0.0** | 0.9 |
| | $\gamma = 3.0$ | 19.4 | 11.2 | **5.6** | 4.5 | 4.2 | **2.3** | **2.0** | **1.4** | 1.0 | 0.5 | 0.9 | **0.0** | 0.9 |
| | $\gamma = 4.0$ | 21.3 | 12.2 | 7.5 | 5.3 | 4.2 | 2.8 | 2.3 | 1.5 | 1.8 | 0.6 | 0.7 | **0.0** | 0.9 |
| | $\gamma = 5.0$ | 27.5 | 16.7 | 10.7 | 7.3 | 5.9 | 3.9 | 3.8 | 1.7 | 1.9 | 0.9 | 0.7 | **0.0** | 0.9 |
| | $\gamma = 6.0$ | 31.9 | 16.7 | 12.3 | 8.1 | 6.0 | 14.5 | 5.1 | 1.9 | 2.1 | 0.9 | 0.7 | 0.1 | 0.9 |
| | $\gamma = 7.0$ | 35.3 | 19.8 | 24.3 | 10.1 | 8.4 | 6.0 | 6.2 | 2.6 | 2.4 | 1.0 | 0.8 | 0.1 | 0.9 |
| | $\gamma = 8.0$ | 39.7 | 23.9 | 25.6 | 11.5 | 8.8 | 12.2 | 7.6 | 3.2 | 2.9 | 1.1 | 0.8 | 0.1 | 0.9 |
| | $\gamma = 9.0$ | 44.4 | 26.1 | 18.2 | 12.9 | 11.0 | 8.5 | 7.9 | 3.5 | 3.4 | 1.1 | 0.8 | 0.1 | 0.9 |
| | $\gamma = 10.0$ | 48.9 | 30.3 | 19.9 | 13.8 | 12.7 | 8.9 | 8.6 | 4.6 | 4.0 | 1.3 | 0.9 | 0.1 | 1.0 |
| SF2 | $\gamma = 0.0$ | 53.7 | 15.4 | 5.0 | 2.4 | 1.9 | 1.3 | **1.1** | 0.2 | 0.2 | 0.1 | **0.0** | 0.3 | 0.0 |
| | $\gamma = 1.0$ | **10.8** | **4.1** | **2.1** | **0.6** | 1.3 | **0.4** | 1.6 | 0.2 | **0.1** | 0.1 | **0.0** | 0.3 | 0.0 |
| | $\gamma = 2.0$ | 11.4 | **4.1** | 2.6 | 0.8 | **1.3** | 0.5 | 1.5 | **0.1** | **0.1** | 0.1 | **0.0** | 0.3 | 0.0 |
| | $\gamma = 3.0$ | 12.2 | 5.4 | 3.0 | 1.4 | 1.7 | **0.4** | 1.4 | **0.1** | **0.1** | 0.0 | **0.0** | 0.3 | 0.0 |
| | $\gamma = 4.0$ | 15.5 | 6.8 | 3.5 | 1.9 | 2.2 | 0.8 | 1.5 | 0.3 | 0.2 | 0.0 | **0.0** | 0.3 | 0.0 |
| | $\gamma = 5.0$ | 18.6 | 8.9 | 4.5 | 2.8 | 2.7 | 1.4 | 1.6 | 0.5 | 0.4 | 0.0 | **0.0** | 0.3 | 0.0 |
| | $\gamma = 6.0$ | 20.8 | 10.9 | 5.9 | 3.3 | 3.5 | 1.8 | 1.8 | 1.0 | 0.5 | 0.0 | **0.0** | 0.3 | 0.0 |
| | $\gamma = 7.0$ | 23.5 | 13.1 | 7.7 | 4.9 | 4.5 | 3.0 | 2.4 | 1.3 | 0.6 | 0.0 | **0.0** | 0.3 | 0.0 |
| | $\gamma = 8.0$ | 27.6 | 14.7 | 9.0 | 5.4 | 6.3 | 3.2 | 3.0 | 1.5 | 0.6 | 0.0 | **0.0** | 0.3 | 0.0 |
| | $\gamma = 9.0$ | 30.1 | 16.9 | 10.7 | 7.0 | 6.8 | 3.7 | 3.5 | 2.0 | 0.9 | 0.0 | **0.0** | 0.3 | 0.0 |
| | $\gamma = 10.0$ | 32.3 | 19.2 | 13.2 | 7.9 | 7.8 | 4.1 | 3.9 | 2.3 | 1.5 | 0.1 | **0.0** | 0.3 | 0.0 |
| ER5 | $\gamma = 0.0$ | 426.5 | 177.1 | 87.8 | 36.9 | 21.5 | 15.7 | 11.2 | 12.3 | 5.7 | 2.6 | **1.9** | 0.1 | **0.0** |
| | $\gamma = 1.0$ | 88.6 | 47.0 | 27.9 | **18.0** | **11.5** | **9.2** | **8.0** | **7.1** | **5.4** | **1.7** | 2.2 | **0.0** | 0.1 |
| | $\gamma = 2.0$ | **73.2** | **44.2** | **26.6** | 20.9 | 13.7 | 10.9 | 10.0 | 8.1 | 6.6 | 2.4 | 2.1 | 0.1 | 0.3 |
| | $\gamma = 3.0$ | 83.6 | 54.4 | 33.7 | 23.4 | 19.0 | 15.0 | 12.4 | 9.7 | 9.1 | 3.3 | 3.5 | 0.2 | 0.4 |
| | $\gamma = 4.0$ | 94.6 | 63.9 | 43.7 | 31.4 | 24.7 | 18.8 | 15.9 | 12.0 | 11.7 | 4.5 | 4.4 | 0.6 | 0.4 |
| | $\gamma = 5.0$ | 110.5 | 75.8 | 51.7 | 39.7 | 31.4 | 24.0 | 19.6 | 16.0 | 13.5 | 6.5 | 4.7 | 1.2 | 0.4 |
| | $\gamma = 6.0$ | 124.9 | 90.0 | 60.2 | 48.0 | 36.4 | 28.7 | 22.7 | 19.1 | 16.5 | 7.9 | 6.2 | 1.3 | 0.4 |
| | $\gamma = 7.0$ | 139.1 | 100.4 | 68.2 | 56.9 | 42.1 | 34.4 | 26.6 | 22.6 | 18.8 | 9.7 | 6.9 | 1.4 | 0.7 |
| | $\gamma = 8.0$ | 152.8 | 110.3 | 77.5 | 65.2 | 53.2 | 39.0 | 31.3 | 25.0 | 23.4 | 11.0 | 7.3 | 1.8 | 0.9 |
| | $\gamma = 9.0$ | 164.8 | 117.9 | 86.0 | 72.0 | 58.0 | 43.3 | 35.1 | 32.1 | 26.8 | 12.7 | 8.1 | 2.4 | 1.4 |
| | $\gamma = 10.0$ | 175.3 | 127.9 | 97.0 | 76.6 | 63.9 | 49.2 | 40.7 | 36.4 | 30.3 | 14.4 | 8.8 | 2.8 | 2.3 |
| SF5 | $\gamma = 0.0$ | 167.9 | 63.9 | 39.1 | **8.5** | 11.5 | **4.0** | **3.2** | **2.0** | 2.1 | **0.3** | 5.0 | **0.0** | 1.9 |
| | $\gamma = 1.0$ | **47.6** | **23.2** | **16.4** | 9.1 | **7.4** | 5.6 | 4.5 | 2.5 | 3.3 | 0.5 | 4.9 | **0.0** | 1.9 |
| | $\gamma = 2.0$ | 58.3 | 37.0 | 25.7 | 17.7 | 12.8 | 12.8 | 7.8 | 6.2 | 5.6 | 1.7 | 5.1 | **0.0** | 1.9 |
| | $\gamma = 3.0$ | 72.0 | 46.1 | 32.1 | 25.4 | 17.2 | 15.6 | 14.2 | 11.6 | 8.9 | 4.1 | 6.1 | **0.0** | 0.1 |
| | $\gamma = 4.0$ | 82.5 | 53.2 | 38.4 | 31.0 | 24.8 | 21.0 | 17.5 | 16.6 | 13.4 | 5.5 | 6.4 | 0.9 | 2.1 |
| | $\gamma = 5.0$ | 91.2 | 62.5 | 45.1 | 36.4 | 29.6 | 24.5 | 22.7 | 19.4 | 15.6 | 8.7 | 7.5 | 1.6 | 2.3 |
| | $\gamma = 6.0$ | 101.7 | 69.6 | 50.5 | 42.2 | 35.0 | 28.2 | 25.4 | 21.3 | 18.0 | 11.9 | 8.2 | 2.4 | 3.4 |
| | $\gamma = 7.0$ | 109.1 | 75.9 | 57.7 | 45.7 | 38.6 | 33.2 | 30.6 | 25.0 | 20.1 | 13.8 | 12.1 | 5.4 | 2.1 |
| | $\gamma = 8.0$ | 116.9 | 82.8 | 64.0 | 50.3 | 41.9 | 36.0 | 33.5 | 27.8 | 21.6 | 14.6 | 14.9 | 6.1 | 2.5 |
| | $\gamma = 9.0$ | 122.2 | 89.1 | 68.7 | 54.2 | 45.0 | 39.6 | 36.3 | 30.4 | 25.2 | 15.8 | 12.1 | 7.8 | 4.5 |
| | $\gamma = 10.0$ | 128.6 | 94.1 | 74.8 | 58.0 | 49.5 | 42.8 | 39.3 | 32.9 | 27.3 | 17.6 | 12.6 | 11.1 | 7.1 |

Table B.10: CDCF-V+ SHD results on different $\gamma$ with different sample size on 100 nodes linear Gumbel SEM.

| | CDCF-V+ | 200 | 300 | 400 | 500 | 600 | 700 | 800 | 900 | 1000 | 1500 | 2000 | 2500 | 3000 |
|---|---|---|---|---|---|---|---|---|---|---|---|---|---|---|
| ER2 | $\gamma = 0.0$ | 124.7 | 35.1 | 11.8 | 6.5 | 3.6 | 2.2 | **0.8** | 0.6 | **0.9** | 0.4 | 0.0 | 0.1 | 0.0 |
| | $\gamma = 1.0$ | 39.8 | 15.6 | 7.3 | 4.0 | 3.0 | 1.5 | 1.2 | **0.5** | 0.9 | 0.4 | 0.0 | 0.1 | 0.0 |
| | $\gamma = 2.0$ | 19.7 | 9.9 | 5.6 | **2.9** | **1.6** | **1.2** | 1.7 | 0.7 | 0.9 | 0.4 | 0.0 | 0.1 | 0.0 |
| | $\gamma = 3.0$ | 15.3 | **7.7** | 4.9 | 3.0 | **1.6** | 1.3 | 1.6 | 0.8 | 1.0 | 0.4 | 0.0 | 0.1 | 0.0 |
| | $\gamma = 4.0$ | **14.7** | 8.8 | **4.7** | 3.4 | 2.0 | 1.5 | 1.7 | 0.9 | 1.3 | 0.5 | 0.0 | 0.1 | 0.0 |
| | $\gamma = 5.0$ | 17.7 | 10.0 | 5.7 | 4.1 | 2.2 | 1.7 | 2.0 | 1.1 | 1.5 | 0.5 | 0.0 | 0.1 | 0.0 |
| | $\gamma = 6.0$ | 19.6 | 12.5 | 6.9 | 4.5 | 3.0 | 2.0 | 2.3 | 1.3 | 1.7 | 0.5 | 0.0 | 0.1 | 0.0 |
| | $\gamma = 7.0$ | 21.9 | 13.8 | 7.8 | 5.7 | 3.0 | 2.7 | 2.4 | 1.5 | 1.7 | 0.5 | 0.1 | 0.1 | 0.0 |
| | $\gamma = 8.0$ | 24.9 | 14.8 | 9.2 | 6.5 | 4.0 | 3.2 | 3.0 | 2.0 | 2.0 | 0.6 | 0.2 | 0.2 | 0.0 |
| | $\gamma = 9.0$ | 27.5 | 16.4 | 9.9 | 7.7 | 4.3 | 3.8 | 3.1 | 2.2 | 2.4 | 0.6 | 0.1 | 0.2 | 0.0 |
| | $\gamma = 10.0$ | 30.9 | 17.6 | 11.2 | 8.7 | 4.5 | 4.2 | 3.3 | 2.9 | 3.1 | 0.8 | 0.1 | 0.2 | 0.0 |
| SF2 | $\gamma = 0.0$ | 46.9 | 8.6 | 1.9 | **0.8** | **0.6** | **0.3** | **0.2** | **0.3** | 0.1 | 0.0 | 0.0 | 0.0 | 0.0 |
| | $\gamma = 1.0$ | 12.0 | 4.5 | **1.5** | **0.8** | **0.6** | **0.3** | **0.2** | 0.4 | 0.1 | 0.0 | 0.0 | 0.0 | 0.0 |
| | $\gamma = 2.0$ | **9.5** | **4.3** | **1.5** | 1.1 | **0.6** | 0.4 | **0.2** | 0.5 | 0.1 | 0.0 | 0.0 | 0.0 | 0.0 |
| | $\gamma = 3.0$ | 9.8 | **4.3** | 1.8 | 1.4 | 1.7 | 0.6 | **0.2** | 0.7 | 0.1 | 0.0 | 0.0 | 0.0 | 0.0 |
| | $\gamma = 4.0$ | 11.9 | 5.2 | 2.1 | 1.8 | 1.8 | 1.1 | **0.2** | 0.7 | 0.1 | 0.0 | 0.0 | 0.0 | 0.0 |
| | $\gamma = 5.0$ | 14.3 | 6.1 | 2.4 | 2.0 | 1.6 | 1.4 | **0.2** | 0.8 | 0.2 | 0.0 | 0.0 | 0.0 | 0.0 |
| | $\gamma = 6.0$ | 15.8 | 7.1 | 3.4 | 2.3 | 2.4 | 1.4 | **0.2** | 0.8 | 0.2 | 0.0 | 0.0 | 0.0 | 0.0 |
| | $\gamma = 7.0$ | 18.5 | 8.0 | 4.9 | 2.8 | 2.5 | 1.7 | 0.6 | 0.9 | 0.2 | 0.0 | 0.0 | 0.0 | 0.0 |
| | $\gamma = 8.0$ | 20.9 | 9.4 | 5.4 | 2.8 | 2.9 | 1.8 | 0.8 | 1.1 | 0.2 | 0.0 | 0.0 | 0.0 | 0.0 |
| | $\gamma = 9.0$ | 22.9 | 11.6 | 6.4 | 3.4 | 3.0 | 2.0 | 1.0 | 1.3 | 0.3 | 0.0 | 0.0 | 0.0 | 0.0 |
| | $\gamma = 10.0$ | 24.3 | 13.3 | 6.6 | 3.8 | 3.3 | 2.5 | 1.3 | 1.4 | 0.4 | 0.0 | 0.0 | 0.0 | 0.0 |
| ER5 | $\gamma = 0.0$ | 362.8 | 139.7 | 70.9 | 40.2 | 23.2 | 17.4 | 9.6 | 5.9 | 6.1 | 1.6 | 0.4 | **0.3** | 0.1 |
| | $\gamma = 1.0$ | 119.1 | 64.7 | 35.9 | 21.6 | 13.3 | 10.7 | **6.8** | 5.5 | **4.7** | 1.5 | 0.5 | **0.3** | 0.1 |
| | $\gamma = 2.0$ | 79.5 | 48.5 | **27.3** | **19.6** | **12.2** | **9.7** | 6.9 | **5.4** | 5.2 | **1.2** | **0.3** | **0.3** | 0.2 |
| | $\gamma = 3.0$ | **74.8** | **46.3** | 29.1 | 20.2 | 12.7 | 11.4 | 8.0 | 6.2 | 6.3 | 1.4 | 0.4 | 0.4 | 0.2 |
| | $\gamma = 4.0$ | 79.6 | 46.8 | 31.9 | 23.3 | 14.9 | 13.0 | 9.3 | 7.9 | 7.0 | 1.5 | 0.7 | 0.5 | 0.3 |
| | $\gamma = 5.0$ | 87.2 | 53.0 | 35.5 | 26.0 | 17.2 | 15.5 | 11.0 | 9.3 | 8.7 | 2.5 | 0.9 | 0.6 | 0.2 |
| | $\gamma = 6.0$ | 99.5 | 60.6 | 39.4 | 31.8 | 20.3 | 17.5 | 14.3 | 10.9 | 10.1 | 2.9 | 1.1 | 0.5 | 0.2 |
| | $\gamma = 7.0$ | 107.4 | 72.0 | 46.0 | 35.7 | 24.9 | 19.7 | 16.7 | 12.5 | 11.8 | 3.5 | 1.5 | 0.5 | 0.2 |
| | $\gamma = 8.0$ | 116.3 | 78.7 | 51.8 | 39.0 | 30.4 | 22.7 | 18.7 | 15.2 | 13.9 | 4.3 | 2.0 | 0.8 | 0.3 |
| | $\gamma = 9.0$ | 124.9 | 85.5 | 57.1 | 43.0 | 33.0 | 26.2 | 20.6 | 16.8 | 15.4 | 5.2 | 2.6 | 0.9 | 0.4 |
| | $\gamma = 10.0$ | 135.3 | 91.9 | 64.5 | 46.8 | 35.9 | 30.1 | 22.9 | 19.5 | 17.5 | 6.0 | 3.3 | 1.0 | 0.5 |
| SF5 | $\gamma = 0.0$ | 104.4 | 31.9 | 12.0 | **7.6** | **5.2** | **2.9** | **2.4** | **2.1** | 2.7 | **0.5** | **0.2** | 0.1 | 0.0 |
| | $\gamma = 1.0$ | **47.2** | **20.2** | **11.0** | 8.1 | 5.8 | 3.6 | 3.3 | 2.2 | **1.9** | **0.5** | 0.3 | 0.1 | 0.0 |
| | $\gamma = 2.0$ | 51.7 | 25.5 | 14.3 | 11.9 | 7.7 | 5.4 | 5.0 | 3.6 | 3.8 | 1.2 | 0.4 | 0.1 | 0.0 |
| | $\gamma = 3.0$ | 58.4 | 30.3 | 20.0 | 14.8 | 10.4 | 8.4 | 7.1 | 5.3 | 5.3 | 1.4 | 0.6 | 0.1 | 0.0 |
| | $\gamma = 4.0$ | 65.9 | 38.1 | 26.5 | 19.9 | 14.9 | 10.2 | 8.9 | 6.5 | 6.0 | 2.4 | 0.9 | 0.3 | 0.0 |
| | $\gamma = 5.0$ | 72.4 | 44.1 | 31.6 | 24.8 | 17.7 | 13.3 | 11.0 | 9.2 | 7.2 | 2.8 | 1.4 | 0.6 | 0.0 |
| | $\gamma = 6.0$ | 78.3 | 49.0 | 37.1 | 27.2 | 20.8 | 15.9 | 14.3 | 10.7 | 9.0 | 3.6 | 1.9 | 0.9 | 0.0 |
| | $\gamma = 7.0$ | 84.8 | 55.2 | 42.1 | 32.8 | 25.0 | 18.3 | 16.1 | 13.3 | 10.1 | 4.5 | 2.7 | 1.5 | 0.3 |
| | $\gamma = 8.0$ | 90.7 | 60.7 | 45.9 | 35.0 | 27.3 | 22.8 | 17.6 | 14.8 | 11.9 | 5.5 | 3.4 | 1.7 | 0.7 |
| | $\gamma = 9.0$ | 97.9 | 64.7 | 49.6 | 38.1 | 31.3 | 25.1 | 19.8 | 16.3 | 14.4 | 6.8 | 3.9 | 2.6 | 0.8 |
| | $\gamma = 10.0$ | 103.4 | 68.2 | 53.3 | 40.8 | 33.9 | 27.5 | 23.8 | 17.9 | 15.4 | 7.7 | 4.4 | 3.3 | 1.4 |

Table B.11: SHD results on 100 nodes linear **monotonous Gaussian noise variance** SEM with different sample size.

| | | 200 | 300 | 400 | 500 | 600 | 700 | 800 | 900 | 1000 | 1500 | 2000 | 2500 | 3000 |
|---|---|---|---|---|---|---|---|---|---|---|---|---|---|---|
| | V | 194.9 | 65.3 | 27.9 | 14.9 | 7.4 | 5.6 | 3.2 | 1.8 | 1.8 | **0.8** | **0.0** | **0.0** | 0.1 |
| | S | 167.4 | 59.6 | 30.5 | 20.9 | 15.9 | 14.6 | 15.1 | 14.5 | 11.8 | 11.6 | 12.0 | 15.7 | 14.3 |
| ER2 | VS | 194.0 | 61.3 | 27.3 | 14.6 | 7.7 | 6.5 | 3.2 | 5.9 | **1.3** | 1.2 | 0.8 | 0.2 | 1.5 |
| | V+ | 99.2 | 38.1 | **16.6** | 11.3 | 5.7 | **4.9** | 3.0 | **1.6** | **1.3** | 0.8 | 0.0 | 0.0 | 0.1 |
| | S+ | **91.0** | 39.2 | 23.5 | 17.8 | 14.6 | 13.5 | 14.4 | 14.3 | 11.8 | 12.0 | 11.4 | 15.8 | 14.3 |
| | VS+ | 101.2 | **36.7** | 18.3 | **10.6** | **5.6** | 5.4 | **2.8** | 6.6 | 2.2 | 1.2 | 0.8 | 0.2 | 1.5 |
| | V | 61.1 | 12.9 | 4.8 | 1.8 | **1.6** | **0.6** | **0.5** | **0.4** | **0.2** | **0.0** | **0.0** | **0.0** | **0.0** |
| | S | 50.6 | 11.6 | 6.3 | 3.9 | 4.0 | 2.4 | 2.4 | 3.3 | 2.7 | 5.1 | 3.1 | 3.5 | 5.4 |
| SF2 | VS | 58.1 | 11.0 | 4.2 | 2.0 | **1.6** | 0.7 | 2.4 | 0.5 | 0.3 | 0.2 | 0.4 | 1.6 | **0.0** |
| | V+ | 30.9 | 9.0 | 3.5 | **1.6** | **1.6** | **0.6** | **0.5** | **0.4** | **0.2** | **0.0** | **0.0** | **0.0** | **0.0** |
| | S+ | **28.5** | 9.3 | 5.2 | 3.8 | 3.9 | 2.4 | 2.4 | 3.3 | 2.7 | 5.1 | 3.1 | 3.5 | 5.4 |
| | VS+ | 29.9 | **8.3** | **3.1** | 1.8 | **1.6** | 0.7 | 2.4 | 0.5 | 0.3 | 0.2 | 0.4 | 1.6 | **0.0** |
| | V | 527.8 | 246.4 | 159.9 | 95.9 | 58.9 | 42.5 | 30.6 | 21.7 | 19.2 | 5.6 | 3.0 | 0.5 | **0.8** |
| | S | 502.4 | 246.8 | 167.3 | 97.1 | 75.8 | 51.4 | 40.0 | 32.8 | 29.7 | 18.2 | 14.9 | 15.3 | 16.3 |
| ER5 | VS | 517.6 | 243.6 | 159.3 | 92.6 | 57.1 | 42.4 | 30.9 | 21.9 | 20.0 | 5.7 | 2.9 | 1.0 | 1.3 |
| | V+ | 252.3 | 141.1 | **95.5** | 61.5 | 43.2 | **30.0** | **21.7** | 16.6 | **15.5** | 4.7 | 2.2 | 0.4 | **0.8** |
| | S+ | 259.3 | 150.0 | 103.8 | 65.5 | 61.5 | 39.4 | 32.9 | 27.2 | 25.5 | 16.5 | 14.8 | 14.8 | 16.3 |
| | VS+ | **250.2** | **135.0** | 96.9 | **58.6** | **41.4** | 31.3 | 21.9 | **16.3** | 16.5 | 4.9 | 2.7 | 0.8 | 1.2 |
| | V | 126.4 | 42.8 | 19.8 | 9.3 | 7.6 | 5.9 | **3.7** | 2.8 | **2.5** | 0.5 | 0.4 | **0.0** | **0.0** |
| | S | 116.5 | 44.9 | 25.9 | 14.9 | 13.8 | 14.3 | 11.9 | 11.1 | 10.4 | 8.7 | 7.7 | 10.0 | 8.3 |
| SF5 | VS | 123.8 | 40.8 | 21.2 | 8.9 | 7.4 | **5.5** | 4.1 | 3.0 | **2.5** | 0.5 | 0.4 | **0.0** | **0.0** |
| | V+ | **70.2** | **28.4** | **14.8** | 8.8 | 6.8 | 5.9 | 3.8 | **2.3** | **2.5** | 0.7 | 0.4 | **0.0** | **0.0** |
| | S+ | 74.1 | 33.9 | 21.1 | 14.0 | 12.9 | 14.6 | 11.8 | 10.5 | 10.5 | 8.9 | 7.7 | 10.2 | 8.3 |
| | VS+ | 70.8 | 28.5 | 17.1 | **8.4** | **6.7** | 5.9 | 3.9 | 2.4 | **2.5** | 0.7 | **0.4** | 0.1 | **0.0** |

Table B.12: SHD results on 100 nodes linear **monotonous Gumbel noise variance** SEM with different sample size.

| | | 200 | 300 | 400 | 500 | 600 | 700 | 800 | 900 | 1000 | 1500 | 2000 | 2500 | 3000 |
|---|---|---|---|---|---|---|---|---|---|---|---|---|---|---|
| | V | 202.5 | 68.8 | 28.3 | 13.9 | 8.6 | 4.7 | 3.5 | 2.0 | 2.0 | 1.0 | **0.0** | 0.2 | 0.0 |
| | S | 172.8 | 65.4 | 28.9 | 18.7 | 15.4 | 18.1 | 14.3 | 11.5 | 15.5 | 16.1 | 13.7 | 14.9 | 16.1 |
| ER2 | VS | 190.5 | 64.7 | 27.4 | 13.6 | 7.5 | 6.6 | 3.7 | 6.2 | **1.6** | 1.0 | 1.2 | 0.3 | 5.8 |
| | V+ | 131.8 | 48.5 | 23.0 | **11.0** | 7.2 | **3.9** | 3.0 | 1.5 | 1.9 | **0.9** | **0.0** | 0.2 | 0.0 |
| | S+ | **123.1** | 52.7 | 23.3 | 17.3 | 15.1 | 17.0 | 14.2 | 11.1 | 14.7 | 16.2 | 13.8 | 14.9 | 16.1 |
| | VS+ | 126.2 | **48.2** | **20.7** | 11.3 | **5.7** | 6.7 | 3.4 | 5.6 | **1.6** | 1.1 | 1.0 | 0.3 | 5.8 |
| | V | 62.4 | 14.0 | 3.5 | 1.6 | 0.9 | 0.8 | 0.7 | **0.5** | **0.2** | **0.0** | **0.0** | **0.0** | **0.0** |
| | S | 53.4 | 12.0 | 4.4 | 4.6 | 4.5 | 2.8 | 2.3 | 2.7 | 2.6 | 4.6 | 3.9 | 2.9 | 3.1 |
| SF2 | VS | 59.8 | 13.6 | 3.3 | **1.5** | **0.8** | **0.6** | 0.8 | 0.6 | 0.4 | **0.0** | **0.0** | **0.0** | **0.0** |
| | V+ | 41.5 | 10.5 | 2.7 | **1.5** | **0.8** | 0.9 | **0.6** | **0.5** | **0.2** | **0.0** | **0.0** | **0.0** | **0.0** |
| | S+ | **37.3** | **9.4** | 3.6 | 4.5 | 4.5 | 3.0 | 2.2 | 2.7 | 2.6 | 4.6 | 3.9 | 2.9 | 3.1 |
| | VS+ | 39.5 | 10.3 | **2.3** | **1.5** | **0.8** | 0.9 | 0.7 | 0.6 | 0.4 | **0.0** | **0.0** | **0.0** | **0.0** |
| | V | 519.8 | 254.2 | 142.4 | 100.4 | 63.9 | 43.3 | 32.8 | 23.7 | 20.7 | 7.3 | 2.8 | 1.3 | **0.5** |
| | S | 501.4 | 246.8 | 152.0 | 107.4 | 77.6 | 52.9 | 40.9 | 33.7 | 35.9 | 20.2 | 17.0 | 12.8 | 14.0 |
| ER5 | VS | 511.6 | 246.0 | 140.9 | 95.6 | 63.6 | 41.6 | 33.1 | 23.0 | 20.1 | 11.4 | 4.7 | 1.4 | 3.3 |
| | V+ | **315.2** | 173.1 | 100.9 | 75.3 | 51.4 | 35.2 | **27.4** | **19.1** | **16.9** | 6.3 | 2.2 | 1.2 | **0.5** |
| | S+ | 318.4 | 175.0 | 112.9 | 83.4 | 66.9 | 42.4 | 36.8 | 28.9 | 34.3 | 19.7 | 17.6 | 12.6 | 14.1 |
| | VS+ | 315.6 | **167.3** | **100.5** | **72.1** | **51.0** | **34.0** | 27.8 | 19.3 | 17.7 | 10.2 | 4.2 | 1.3 | 3.3 |
| | V | 137.5 | 49.4 | 18.7 | **10.1** | **6.7** | 4.9 | 3.8 | 2.3 | 1.9 | 0.8 | 0.2 | 0.1 | 0.0 |
| | S | 128.1 | 47.1 | 25.7 | 15.9 | 16.4 | 11.5 | 11.3 | 10.1 | 8.5 | 10.1 | 9.2 | 9.1 | 12.5 |
| SF5 | VS | 132.5 | 44.2 | 19.2 | 10.3 | 8.7 | 5.6 | 4.1 | 2.5 | **1.9** | 0.8 | **0.2** | **0.1** | **0.0** |
| | V+ | 89.7 | 33.1 | **15.4** | **10.1** | 6.8 | 5.0 | 4.0 | 2.4 | **1.9** | 0.7 | **0.2** | **0.1** | **0.0** |
| | S+ | 89.3 | 37.1 | 22.6 | 15.9 | 16.9 | 11.5 | 11.6 | 10.2 | 8.5 | 10.0 | 9.2 | 9.0 | 12.5 |
| | VS+ | **85.0** | **32.2** | 15.5 | 11.3 | 8.5 | 5.8 | 4.4 | 2.6 | **1.9** | **0.7** | **0.2** | **0.1** | **0.0** |

Table B.13: SHD results on 100 nodes linear **monotonous Exponential noise variance** SEM with different sample size.

| | | 200 | 300 | 400 | 500 | 600 | 700 | 800 | 900 | 1000 | 1500 | 2000 | 2500 | 3000 |
|---|---|---|---|---|---|---|---|---|---|---|---|---|---|---|
| ER2 | V | 210.8 | 67.0 | 31.6 | 13.2 | 11.2 | 6.3 | 3.4 | 2.9 | 1.9 | **0.3** | **0.2** | **0.2** | **0.0** |
| | S | 182.6 | 63.5 | 32.8 | 21.4 | 16.0 | 17.0 | 11.1 | 13.2 | 12.1 | 14.6 | 11.1 | 16.9 | 16.7 |
| | VS | 202.8 | 90.2 | 30.8 | 12.5 | 9.9 | 5.2 | 4.8 | 5.7 | 2.3 | 0.5 | 1.2 | 1.3 | 0.2 |
| | V+ | 108.7 | 42.1 | 20.0 | 9.0 | 7.1 | 4.5 | **3.0** | **2.4** | **1.8** | **0.3** | **0.2** | **0.2** | **0.0** |
| | S+ | **100.6** | **41.6** | 22.8 | 17.0 | 13.5 | 16.1 | 11.7 | 12.7 | 11.8 | 14.4 | 11.0 | 16.7 | 16.7 |
| | VS+ | 107.3 | 58.4 | **19.1** | **8.3** | **6.9** | **4.3** | 3.3 | 5.2 | 2.1 | 0.4 | 1.2 | 1.3 | 0.2 |
| SF2 | V | 71.4 | 13.0 | 6.2 | 1.7 | **0.8** | 0.9 | **0.7** | **0.2** | 0.6 | 0.3 | 0.0 | 0.0 | 0.0 |
| | S | 61.2 | 14.4 | 4.7 | 4.7 | 2.9 | 3.3 | 4.0 | 4.6 | 2.2 | 3.0 | 2.8 | 3.8 | 5.3 |
| | VS | 76.0 | 14.1 | 3.5 | 2.7 | **0.8** | 0.6 | 0.8 | 0.5 | **0.6** | **0.3** | 0.0 | 0.9 | 0.0 |
| | V+ | 38.9 | 8.7 | 3.7 | **1.6** | 1.0 | 0.7 | **0.7** | **0.2** | 0.6 | 0.3 | 0.0 | 0.0 | 0.0 |
| | S+ | **37.4** | 10.0 | 4.4 | 4.6 | 3.2 | 4.1 | 4.0 | 4.6 | 2.2 | 3.0 | 2.8 | 3.8 | 5.3 |
| | VS+ | 43.3 | **8.2** | **2.9** | 2.4 | 1.0 | **0.5** | 0.8 | 0.5 | **0.6** | **0.3** | 0.0 | 0.9 | 0.0 |
| ER5 | V | 563.5 | 267.7 | 149.4 | 97.5 | 67.9 | 45.5 | 33.1 | 23.7 | 18.8 | 6.8 | 2.0 | 1.5 | 0.5 |
| | S | 529.5 | 258.8 | 153.8 | 105.1 | 79.8 | 58.6 | 47.4 | 31.5 | 32.2 | 23.4 | 17.0 | 13.9 | 14.9 |
| | VS | 549.0 | 258.3 | 151.0 | 97.0 | 68.4 | 46.1 | 35.4 | 22.9 | 18.5 | 7.7 | 4.9 | 2.8 | 0.9 |
| | V+ | 268.1 | 150.9 | **90.3** | **61.6** | 48.9 | **32.3** | **23.5** | 19.1 | **14.0** | **5.6** | **1.6** | **1.1** | **0.4** |
| | S+ | 262.6 | 156.3 | 99.8 | 73.0 | 60.9 | 45.1 | 39.7 | 26.4 | 26.3 | 22.2 | 16.7 | 15.0 | 14.7 |
| | VS+ | **258.7** | **146.1** | 95.2 | 63.9 | **48.7** | **32.3** | 27.2 | **17.9** | 14.4 | 7.3 | 3.8 | 2.5 | 1.0 |
| SF5 | V | 147.4 | 45.6 | 29.8 | 11.2 | 11.2 | 4.8 | **3.7** | **2.7** | **2.0** | **1.2** | 0.5 | **0.0** | **0.0** |
| | S | 122.3 | 48.5 | 26.6 | 16.5 | 12.9 | 12.4 | 12.5 | 10.0 | 12.5 | 12.0 | 13.1 | 8.6 | 13.5 |
| | VS | 136.9 | 43.9 | 20.6 | 11.3 | **6.4** | 4.9 | 5.0 | 2.9 | 2.6 | **1.2** | 0.5 | 3.6 | 0.1 |
| | V+ | 77.7 | 32.2 | 19.3 | **8.7** | 6.6 | **4.7** | 4.1 | **2.7** | 2.5 | 1.4 | **0.3** | **0.0** | **0.0** |
| | S+ | **74.3** | 37.8 | 22.1 | 15.7 | 13.1 | 12.0 | 12.7 | 10.2 | 12.9 | 12.3 | 13.1 | 8.6 | 13.5 |
| | VS+ | 78.2 | **31.5** | **17.1** | 10.2 | 6.6 | 5.9 | 5.2 | 3.1 | 3.2 | 1.4 | **0.3** | 3.6 | 0.1 |

