# OpenReview forum: "Causal Discovery via Cholesky Factorization"
_ICLR.cc/2022/Conference — ICLR 2022 Submitted_

### Official Review · Reviewer_WtL4 · 2021-11-02

**Correctness:** 3
**Technical Novelty And Significance:** 3
**Empirical Novelty And Significance:** 3
**Recommendation:** 6
**Confidence:** 3

**Main Review:**

The paper presents a new algorithm to discover the causal relationship in a directed acyclic graph. The authors restrict themselves to the linear SEM setting under the assumption that the conditional noise variance for child nodes is larger than that of the parent nodes which ensures that the problem is identifiable.
Overall I believe that this is a good paper, but I need a few clarifications.
1. The authors say that the conditional noise variance of child nodes is "approximately" larger than that of the parent nodes. What approximation are the authors talking about?
2. While evaluating the algorithm, the authors generate ER graphs. How do the authors ensure that the resulting graphs are DAGs?

**Summary Of The Paper:**

In this paper the authors consider the problem of learning directed graphical models in the linear SEM setting. The authors use iterative Cholesky factorization of the covariance matrix in order to learn the causal relationship from the data.

**Summary Of The Review:**

Overall an efficient algorithm to learn causal relationship. The paper is held back by strong assumptions.

---

### Official Review · Reviewer_Ag69 · 2021-11-02

**Correctness:** 1
**Technical Novelty And Significance:** 3
**Empirical Novelty And Significance:** Not applicable
**Recommendation:** 3
**Confidence:** 4

**Main Review:**

Strength : The paper provides thorough discussion of previous studies.
Weak: Some points of the paper are not clear.
(1)	The new identifiability condition (V) on page 4 requires a clear justification. For example, suppose that a true graph is V = (1,2,3) and E = { (2,3) }. If, node 1is chosen as i_1. Then, U_{k-1} y_j = 0 for j = 2, 3. Hence, I think the condition (V) and (SV) require a mathematical proof.
(2)	The require conditions for Theorem 2.1 appear to be unrealistic. The first condition is that bounded random vector \| x\|^2 \leq R. In this case, it is definitely not satisfied when error variables are Gaussian and Exponential as applied in the numerical experiments. Furthermore, as the number of node increases, R increases. In addition, R clearly depends on the sparsity level. For instance, X1 ~ N(0, \sigma^2), X2 = X1 + N(0, \sigma^2), X3 = X1 + X2 + N(0, \sigma^2). Then, E( X1^2 + X2^2 + X3^2 ) = \sigma^2 + 2 \sigma^2 + 4 \sigma^2 = 7 \sigma^2. However, for a sparser case, X1 ~ N(0, \sigma^2), X2 = X1 + N(0, \sigma^2), X3 = X1 + N(0, \sigma^2), E( X1^2 + X2^2 + X3^2 ) = \sigma^2 + 2 \sigma^2 + 2 \sigma^2 = 5 \sigma^2.
(3)	The numerical experiments do not support the theoretical findings of the paper. According to Theorem 2.1., the sample complexity does not depend on the sparsity level of a graph. Hence, the comparison of performance of the proposed method for learning ER2 and ER5 cannot be explained. In addition, it is unclear that the SHD converges to zero. Hence, it would be better to provide the empirical probability of P(TRIU(U_p) = T).
(4)	As explained, the considered distributions (Gaussian, Exponential) do not satisfy the bounded random variable assumption.


**Summary Of The Paper:**

This paper develops a new algorithm for learning linear structural equation models using cholesky factorization. This paper explains that the proposed algorithm is consistent in high dimensional settings and computational feasible. This paper thoroughly discusses the recent studies of learning linear SEMs and provide a clear motivation. Furthermore, this paper provides a lot of numerical experiments to support its theoretical findings.


**Summary Of The Review:**

This paper has a good idea of learning linear SEMs. However, this version of paper includes some unrefined conditions and requires better simulation settings.

I believe that this paper has a great potential and it would be a much better paper after modifying some points of the paper.

---

### Official Review · Reviewer_2cwa · 2021-11-02

**Correctness:** 4
**Technical Novelty And Significance:** 2
**Empirical Novelty And Significance:** 2
**Recommendation:** 6
**Confidence:** 2

**Main Review:**

The paper is well written and the claims are well supported by the experiments and theoretical analysis of the method. The significance of the work is on the time-complexity side, which is better than the existed related works. The correctness and soundness of the method are shown by the theoretical analysis and the experimental results. My only concern is that the work is based on a quite restrictive class of SEMs, of which the extension to the nonlinear cases or the more general scenario is not so clear.

**Summary Of The Paper:**

The paper works on causal discovery in the linear Gaussian case, on which the identifiability is based on (Peters & Bühlmann, 2014). The proposed method is based on Cholesky factorization and has better efficiency/time-complexity performance than the related state-of-art methods. Moreover, it also provides a theoretical analysis of the resulted graph, which is appreciated. The experiments can support the claims.

**Summary Of The Review:**

The paper is well written and the contributions and claims are well supported, while the proposed method focuses on a restrictive class of SEMs.

---

### Official Review · Reviewer_tZir · 2021-11-02

**Correctness:** 3
**Technical Novelty And Significance:** 2
**Empirical Novelty And Significance:** Not applicable
**Recommendation:** 3
**Confidence:** 5

**Main Review:**

The present work uses several techniques from linear model causal discovery. However, the relation to existing work is not made clear enough in my opinion. In particular, how different is the proposed method (at least the ordering part since the pruning procedure, which is done by thresholding the estimated precision matrix, also appears in [2]) from the ordering estimation using the forward stepwise section ([1], Algorithm 1) ?

On the theoretical side, the claimed improved sample complexity compared to existing results seems unfair. Indeed, Theorem 2.1 assumes that the support of the data distribution lies within a sphere of radius sqrt(R), for a constant R independent of the dimension. Table 2.1 compares this result to the one stated in [2], which only assumes that the noise involved in each equation of the SEM is sub-Gaussian, e.g., each noise is a standard Gaussian. In this scenario (which appears to be tried in the experiments), not only the boundedness assumption does not hold, but the expected value of ||x||^2 would scale at least linearly in the dimension, and even quadratically for dense graphs, involving additional dimension dependence in the sample complexity.

The required identifiability assumption (Theorem 2.1) relies on an estimate from the data, not the model parameters themselves, which is strange to me, and differs from the conditions provided in [1,2] as opposed to what is claimed.

On the practical side:
What part of the algo is actually improved compared to existing order-search algorithms ? Finding the topological order ? By better estimating the covariance matrix ? Using regularisation ? Pruning the DAG ?

You mention that algorithms such as LISTEN or US perform worse (or equally) compared to the the selected baselines. I find this fact very surprising, since the corresponding papers mention much better performance than what is obtained with, e.g., NOTEARS. It would also be interesting to compare your method with LISTEN using the inverse of the empirical covariance matrix (as opposed to the CLIME estimator), so as to match the covariance estimation part more closely with yours.

[1] Park, "Identifiability of Additive Noise Models Using Conditional Variances"
[2] Ghoshal et al., "Learning linear structural equation models in polynomial time and sample complexity"

**Summary Of The Paper:**

This paper proposes a method for recovering the causal graph of additive linear models from purely observational data, under some an identifiability assumption, that seems to be related to the forward step-size assumption of [1]. Their algorithm is based on iteratively identifying a root of the causal graph based on its conditional variance. Once a topological order is learned, the graph is constructed by thresholding the Cholesky factor of the permuted precision matrix. The proposed algorithm is then tested and compared on both synthetic and real-world data.

**Summary Of The Review:**

I find that the improved theoretical guarantees are not well motivated, and comparison to existing algorithms not clear enough.

---

### Official Review · Reviewer_SYfm · 2021-11-08

**Correctness:** 4
**Technical Novelty And Significance:** 3
**Empirical Novelty And Significance:** 3
**Recommendation:** 3
**Confidence:** 5

**Main Review:**

The paper is well motivated and the contributions of the paper is clear — the authors develop the fastest algorithm for learning linear SEMs. However given that existing algorithms from Ghoshal and Honorio 2018 and Chen et al 2019 are already polynomial time and are comparable to the proposed algorithm in terms of running time I feel this is not significant. The paper doesn't contribute anything new towards identifiability of linear SEMs. The experiments are extensive and the clearly demonstrate the effectiveness of the method.

However my main criticism is the sample complexity. The sample complexity depends on the $\ell_2$ norm of the $p$ dimensional random vectors $X$ which can be $O(p)$ even when the underlying SEM is sparse (has degree $d$ that is constant). This is not the case with the algorithm proposed in Ghoshal and Honorio whose sample complexity grows as $O(\mathrm{poly}(d) \log p)$. So the improved computational complexity come at the cost of increased sample complexity and the algorithm is not applicable in high dimensional settings.

In the experiments, the authors compare the performance of their algorithm against LISTEN (Ghoshal and Honorio 2018) only at 3000 number of samples. I would have liked to see learning curves (SHD vs varying number of samples) for both the algorithms. It is likely that LISTEN outperforms the proposed algorithm when the number of samples is low.

**Summary Of The Paper:**

The paper proposes a new algorithm for learning linear SEMs using Cholesky factorization of the covariance matrix induced by the SEM. While the paper essentially follows the ideas in Ghoshal and Honorio 2018 and Chen et al 2019, the main innovation is combining the order search step followed by parent set recovery into a single step through Cholesky factorization.

**Summary Of The Review:**

While the proposed algorithm improves upon existing algorithms for learning linear SEMs, it comes at the cost of increased sample complexity. Unlike existing algorithms, the proposed algorithm is not applicable in the high-dimensional regime since the sample complexity grows as $O(p \log p)$. Given that existing algorithms are already polynomial time and the proposed algorithm doesn't improve upon them in terms of required identifiability conditions and has significantly worse sample complexity, I feel the contributions are not strong enough.

---

### Decision · Program_Chairs · 2022-01-20

**Decision:**

Reject

**Comment:**

This paper proposes an algorithm for learning linear SEMs via the Cholesky factorization and provides a detailed theoretical analysis of the algorithm. After an extensive discussion and clarification from the authors, there was a consensus that the theoretical results are incremental compared to existing work and many of the claims need additional context in light of existing work. In particular, I recommend the authors pay careful attention to the presentation of the sample complexity bounds, which were revealed to be substantially weaker than initially claimed, and to validate these bounds with careful experiments.